

# UK Global Ocean GO6 and GO7: a traceable hierarchy of model resolutions

David Storkey[1], Adam T. Blaker[2], Pierre Mathiot[3,1], Alex Megann[2], Yevgeny Aksenov[2], Edward W. Blockley[1], Daley Calvert[1], Tim Graham[1], Helene T. Hewitt[1], Patrick Hyder[1], Till Kuhlbrodt[4], Jamie G.L. Rae[1], and Bablu Sinha[2]

[1]Met Office, FitzRoy Road, Exeter EX1 3LX, UK
[2]Marine Systems Modelling, National Oceanography Centre, Southampton SO14 3ZH, UK
[3]British Antarctic Survey, High Cross, Madingley Road, Cambridge CB3 0ET, UK
[4]National Centre for Atmospheric Science, Department of Meteorology, University of Reading, Reading RG6 6BB

*Correspondence to:* David Storkey (dave.storkey@metoffice.gov.uk)

**Abstract.** Versions 6 and 7 of the UK Global Ocean configuration (known as GO6 and GO7) will form the ocean components of the Met Office GC3.1 coupled model and UKESM earth system model to be used in CMIP6 simulations. The label "GO6" refers to a traceable hierarchy of three model configurations at nominal $1°$, $1/4°$ and $1/12°$ resolution. The GO6 configurations are described in detail with particular focus on aspects which have been updated since the previous version (GO5). Results

of 30-year forced ocean-ice integrations with the $1/4°$ model are presented, in which GO6 is coupled to the GSI8.1 sea ice configuration and forced with CORE2 fluxes. GO6-GSI8.1 shows an overall improved simulation compared to GO5-GSI5.0, especially in the Southern Ocean where there are more realistic summertime mixed layer depths, a reduced near-surface warm and saline biases and an improved simulation of sea ice. The main drivers of the improvements in the Southern Ocean simulation are tunings of the vertical and isopycnal mixing parameters. Selected results from the full hierarchy of three

resolutions are shown. Although the same forcing is applied, the three models show large-scale differences in the near-surface circulation and in the short-term adjustment of the overturning circulation. The GO7 configuration is identical to the GO6 $1/4°$ configuration except that the cavities under the ice shelves are opened. Opening the ice shelf cavities has a local impact on temperature and salinity biases on the Antarctic shelf with some improvement in the biases in the Weddell Sea.

## 1 Introduction

Since 2010 the UK Met Office, the National Oceanography Centre and the British Antarctic Survey have collaborated on the development of standard global ocean model configurations based on the NEMO code (Madec, 2016). These are intended to be used for a variety of applications across a range of timescales from ocean forecasting a few days ahead to century-scale climate modelling. The use of a single ocean model configuration for multiple applications is in the spirit of the seamless forecasting approach (Brown et al., 2012).

This paper describes the latest Global Ocean configurations GO6 and GO7, and presents results of testing them in forced mode with the GSI8.1 configuration of the CICE sea ice model. The GO6 ocean model is the ocean component of the GC3.1



version of the Met Office Hadley Centre coupled climate model (Williams et al. (submitted 2017), Hewitt et al. (2016)) and the ocean component of the UKESM1 UK Earth System model (Kuhlbrodt et al., in preparation), both of which will be used in CMIP6 simulations (Eyring et al., 2016) and associated OMIP simulations (Griffies et al., 2016) . GO6 is expected to be incorporated into future versions of the FOAM ocean forecasting system (Blockley et al., 2014), the GloSea seasonal

forecasting system (MacLachlan et al., 2015) and the DePreSys decadal forecasting system (Dunstone et al., 2016).

The previous configuration GO5 (Megann et al., 2014) was only released at a single resolution of a nominal $1/4°$ horizontal grid spacing. GO6 is a traceable hierarchy of three horizontal resolutions: $1°$, $1/4°$ and $1/12°$, all with the same vertical grid. By traceable we mean that the only differences between the three configurations are those that can be justified as necessitated by the change in resolution, an example being tuning of the horizontal viscosity. The main focus of this paper is on the $1/4°$

configuration, but we also present selected results from the traceable hierarchy of resolutions in Section 6. GO7 is identical to GO6 except that the ice shelf cavities are open. GO7 currently only exists at $1/4°$ resolution.

The paper is structured as follows. Section 2 gives a full model description of GO6; Section 3 describes the initialisation and forcing data sets for the forced tests; Section 4 describes the aspects of the model that have changed relative to the GO5 configuration and presents some results of sensitivity experiments; Section 5 presents an overall model evaluation of the $1/4°$

model comparing GO6 to GO5; Section 6 shows results from the hierarchy of resolutions at GO6; Section 7 briefly describes the impact of opening the cavities under the ice shelves and Section 8 presents a summary and indication of future developments.

## 2   Model description

GO6 and GO7 are part of the Global Ocean (GO) configuration series, building on the GO5 model described by Megann et al. (2014). This section provides a full model description of GO6/GO7 and also a brief overview of the GSI8.1 sea ice model. GO6

and GO7 are based on version 3.6 of the NEMO ocean model code (Madec, 2016). The $1/4°$ and $1/12°$ models are closely related to the corresponding global DRAKKAR configurations (Barnier et al. (2006) and Drakkar (2017)) sharing many of the same dynamics and physics choices.

### 2.1   Model grid and bathymetry

The hierarchy of resolutions is based on the "ORCA" family of global grids within the NEMO framework (Madec, 2016),

specifically ORCA1, ORCA025 and ORCA12. These have nominal $1°$, $1/4°$ and $1/12°$ resolution at the equator and an isotropic Mercator grid in which the meridional grid spacing is reduced to match the reduction in the zonal grid spacing in the poleward direction. In the northern hemisphere there is a quasi-isotropic bipolar grid with poles at land points in Siberia and Canada. The ORCA1 model has increased resolution in the meridional direction near the equator. In the southern hemisphere, the grids have been extended southwards compared to the original versions[1] with the southern limit changed from $77°S$ to $85°S$

to permit the modelling of the circulation under ice shelves in Antarctica. A simple extension of the Mercator grid southwards would result in very small grid spacings at the southernmost points giving a severe CFL limit on the timestep. An alternative

---

[1]The extended versions are sometimes referred to as eORCA1, eORCA025 and eORCA12



procedure described in Mathiot et al. (2017) is therefore used, whereby segments of the northern hemisphere bipolar part of the grid are scaled and joined to the southern edge of the existing grid.

The three models share a common set of 75 vertical levels (Culverwell, 2009). The level thickness is 1m near the surface and 200m at depth, increasing as a double tanh function. This gives a balance between high resolution near the surface to resolve
short-term ocean responses to atmospheric forcing and reasonable resolution in the thermocline. Stewart et al. (2017) show that 75 levels is the minimum number capable of resolving the second baroclinic mode. Cells spanning partial model levels are allowed next to the bathymetry (Barnier et al. (2006), Adcroft et al. (1997)).

The model bathymetries for each of the three resolutions were derived from different data sets; in this respect the hierarchy of resolutions is not yet fully traceable. The ORCA1 bathymetry is derived from the ETOPO2 dataset (NOAA, 2006) with
the bathymetry on the Antarctic shelf based on IBSCO (Arndt et al., 2013). For ORCA025, the bathymetry is derived from the ETOPO1 dataset (Amante and Eakins, 2009) with modifications in coastal regions based on GEBCO (IOC, 2008) and the bathymetry on the Antarctic shelf derived from IBSCO (Arndt et al., 2013). For ORCA12 the bathymetry is derived from GEBCO_2014 (Weatherall et al., 2015).

### 2.2 Free surface solution and advection

The model uses a non-linear free surface in which the cell thicknesses throughout the water column are allowed to vary with time (the $z^*$ coordinate of Adcroft and Campin (2004)). This permits an exact representation of the surface fresh water flux. The equation for the surface pressure gradient is solved using a filtered solution in which the fast gravity waves are damped by an additional force in the equation (Roullet and Madec 2000).

The momentum advection term is a vector-invariant formulation in which the horizontal advection is split into rotational
and irrotational parts. The vorticity term (including the Coriolis term) is calculated using the energy and enstrophy conserving scheme of Arakawa and Lamb (1981). There are two versions of this scheme in NEMO which differ according to how the topography boundary condition is handled. For GO6 we choose the *ln_dynvor_een_old=true* option which reinforces the tendency of the flow to follow isobaths (Madec (2016) Section 6.2). This is the same option that was used for GO5. The irrotational part of the momentum advection is formulated according to Hollingsworth et al. (1983) in order to avoid vertical
numerical instabilities. Advection of tracers is done using the Total Variance Dissipation (TVD) scheme of Zalesak (1979).

### 2.3 Mixing and boundary conditions

Lateral diffusion of momentum is on geopotential surfaces and uses a laplacian viscosity in ORCA1 and a bilaplacian viscosity in ORCA025 and ORCA12 with coefficients given in Table 1. The viscosity coefficients reduce polewards - linearly with grid size for the laplacian and with the cube of the grid size for the bilaplacian - in order to avoid numerical diffusion
instabilities. Lateral diffusion of tracers is along isoneutral surfaces using laplacian mixing with coefficients given in Table 1. A parameterisation of adiabatic eddy mixing based on the local stratification (Held and Larichev, 1996) is used in ORCA1 but not in ORCA025 or ORCA12. A free slip lateral boundary condition on momentum is applied at all resolutions. For the $1/4°$ and $1/12°$ configurations, the momentum boundary condition is changed around the coastline of Antarctica to a partial slip





condition at $1/4°$ and a no-slip condition at $1/12°$. This is to avoid instabilities associated with artificial cliffs in the bathymetry at the edge of the ice shelves where the ice cavities have been closed.

The vertical mixing of tracers and momentum is parameterised using a modified version of the Gaspar et al. (1990) turbulent kinetic energy (TKE) scheme (Madec, 2016). Unresolved mixing due to internal wave breaking is represented by a background

vertical eddy diffusivity of $1.2 \times 10^{-5}$ m$^2$/s, which decreases linearly from $\pm 15°$ latitude to a value of $1.2 \times 10^{-6}$ m$^2$/s at $\pm 5°$ latitude (Gregg et al. 2003) and a globally constant background viscosity of $1.2 \times 10^{-4}$ m$^2$/s. Additionally there is enhanced mixing at the surface depending on the wind stress to represent mixing due to surface wave breaking (Craig and Banner, 1994), a representation of Langmuir cell mixing (Axell, 2002), and an ad hoc representation of mixing due to near-inertial wave breaking (Rodgers et al., 2014). This latter term has an associated length scale which can be varied geographically. This was

tuned at GO5 (Megann et al., 2014) and further tuning of this term has taken place as part of the development of GO6, which is described in Section 4.3.

Convection in the model is parameterised as an enhanced vertical diffusivity of 10 m$^2$/s for momentum and tracer fields where the water column is unstable. A parameterisation of double diffusive mixing is included (Merryfield et al., 1999). A climatological geothermal heat flux due to (Stein and Stein, 1992) is used as a bottom boundary condition. A quadratic

bottom friction is used with increased friction in the Indonesian Throughflow, Denmark Strait and Bab el Mandeb regions. The bottom boundary layer scheme of (Beckmann and Döscher, 1997) is used with advective and diffusive components. The tidal mixing parameterisation of Simmons et al. (2004) is included with a special formulation for the Indonesian Throughflow (Koch-Larrouy et al., 2008).

### 2.4 Fresh water input from land

Fresh water runoff from land is input in the surface layer of the ocean with the assumption that the runoff is fresh and at the same temperature as the local sea surface temperature. An enhanced vertical mixing of $2 \times 10^{-3}$ m$^2$/s is added over the top 10 m of the water column at runoff points to mix the runoff vertically and avoid instabilities associated with very shallow fresh layers at the surface. Fresh water input from ice sheets in Greenland and Antarctica is modelled using a Lagrangian iceberg scheme and a parametrisation of ice shelf basal melting .

The Lagrangian iceberg model is that of Bigg et al. (1997) and Martin and Adcroft (2010), which was implemented in NEMO by Marsh et al. (2015). Icebergs are represented by Lagrangian particles with each particle representing a collection of icebergs within a given size range. The momentum balance for icebergs comprises the Coriolis force, air and water form drags, the horizontal pressure gradient force and a wave radiation force. The mass balance for an individual iceberg is governed by basal melting, buoyant convection at the sidewalls, and wave erosion. While this model gives a reasonable approximation

of iceberg behaviour, there are a number of weaknesses. Firstly, the icebergs only exchange heat and fresh water with the surface layer of the ocean (of 1m thickness in GO6) and are advected by surface currents, whereas in reality many icebergs have draughts of hundreds of metres and so will interact with the ocean at depth. Secondly, there is no momentum exchange with sea ice. Thirdly, while the latent heat of melting is extracted from the ocean, the heat content of the meltwater input to the ocean is neglected.



The extended versions of the ORCA grids and developments in the NEMO 3.6 code make it possible to model the ocean circulation beneath the major ice shelves around Antarctica (Mathiot et al., 2017). In the GO6 configuration we close these cavities and prescribe climatological fresh water input through depth at the edge of the ice shelves to mimic the effect of ice shelf basal melt on the wider circulation, a parametrisation described in Mathiot et al. (2017). In GO7 the cavities are open and

climatological melting is prescribed at the bottom of the ice shelf.

### 2.5   Sea ice model

For the CORE2-forced experiments described in this paper GO6 and GO7 are run with the GSI8.1 configuration of the Los Alamos National Laboratory sea ice model CICE. The GSI8.1 configuration is described in detail by Ridley et al. (2017). It consists of version 5.2.1 of the CICE base code with multi-layer, energy-conserving thermodynamics (Bitz and Lipscomb,

1999), elastic-viscous-plastic ice rheology (Hunke and Dukowicz, 1997) and multi-category ice thickness (Bitz et al., 2001) with 5 thickness categories. The multi-layer thermodynamics uses four layers of ice and one layer of snow. The impact of surface melt ponds is included, with the melt pond faction and depth calculated using the topographic melt pond model of Flocco et al. (2010). The freezing point of the ocean is dependent on the local salinity. Ridley et al. (2017) describe tests of GSI8.1 within the context of the GC3.1 coupled model (Williams et al., submitted 2017). In the coupled configuration the sea

ice thermodynamics calculation is split, with the internal sea ice thermodynamics being solved in CICE, but the surface energy balance is solved in the land surface model JULES as described by West et al. (2016). For the forced experiments described in this paper, the sea ice thermodynamics are solved entirely within CICE.

While the ocean model is solved on the Arakawa C-grid (Arakawa, 1966)), the sea ice model on the Arakawa B-grid, with the tracer points of the two grids aligned with each other. An interpolation routine is used to couple ocean and ice velocities. To

avoid issues related to the difference in model grids, single point inlets were filled in around Antarctica in all three resolutions. As in the coupled model, the ice and ocean components are combined into a single executable, avoiding the need for a coupler.

### 3   Forcing datasets and initial conditions

For the forced experiments described in this paper, GO6 was driven over the period 1976–2005 by the CORE2 surface forcing data set (Large and Yeager, 2009). The bulk formulae proposed by Large and Yeager (2004) is used to calculate turbulent flux

transfer coefficients. Wind stresses are calculated as the relative stress between the wind and the ocean current. An artificial diurnal cycle is imposed on the daily mean shortwave fluxes such that the total incident energy is unchanged.

A restoring fresh water flux is applied to restore the sea surface salinity (SSS) towards monthly mean climatological values. In common with many global ocean models this is necessary to avoid large drifts in the salinity and overturning circulation. The retroaction coefficient used is $-33.333$ mm/day/psu, corresponding to a piston velocity of about 50 m / 4 years which

Griffies et al. (2009) describe as weak restoring.

A climatological monthly runoff field derived by Bourdalle-Badie and Treguier (2006) from the Dai and Trenberth (2002) climatology is applied. The Lagrangian iceberg scheme is fed with a seasonal climatology of glacial accumulation at the shore





due to Marsh et al. (2015). The parametrisation of ice shelf basal melt around Antarctica uses climatological data from Rignot et al. (2013).

Initial conditions for temperature and salinity are obtained from monthly climatologies based on the "EN" reanalyses. For the comparison of GO6 and GO5 on the $1/4°$ grid in Section 5 the initial conditions are from a climatology based on the
EN3 monthly objective analysis (Ingleby and Huddleston, 2007), years 2004-2008. For other results presented in this paper the initial conditions were from a climatology based on the years 1995-2014 from the more recent EN4 objective analysis (Good et al., 2013). In all cases the model is spun up from a from a state of rest. The initial conditions for the sea ice were taken from a time-mean of 20 years of January-mean fields from of a present-day forcing integration of the GC2 coupled model (Williams et al., 2015).

## 4  Developments since GO5 and sensitivities

In this section, we describe in more detail the model changes between GO5 and GO6/GO7 and present results of sensitivity experiments showing the impact of individual changes. The integrations for sensitivity tests were often shorter than the 30-year test performed for the GO6-GO5 comparison; details for each set of results are noted in the figure captions.

### 4.1  NEMO version upgrade and nonlinear free surface

The version of the NEMO base code has been upgraded from NEMO 3.4 to NEMO 3.6 which was released in 2015. Among the new features available at NEMO 3.6 (compared to 3.4) are the Hollingsworth et al. (1983) formulation for momentum advection, the Lagrangian icebergs scheme and the ability to simulate the circulation beneath ice shelves. The first two of these are activated in GO6 and are discussed in Sections 4.2 and 4.5. Testing of the under-ice-shelf functionality is described in Section 7.

GO5 employed a linear free surface in which the volume of the ocean remains constant and the surface fresh water flux is represented as a virtual salt flux (Roullet and Madec, 2000). At GO6 we switch to the nonlinear free surface in which the volume of the ocean grid cells throughout the water column is allowed to vary with time (the $z^*$ coordinate of Adcroft and Campin (2004)). This has the advantage that the fresh water flux at the surface can be represented accurately. It is also a prerequisite for the future inclusion of tides in the global model.

Both the upgrade to the base code version and the switch to the nonlinear free surface have been tested in separate sensitivity experiments and have been shown to have small impacts on the large-scale simulated climate (not shown).

### 4.2  Improved formulation of momentum advection

Hollingsworth et al. (1983) describe a computational symmetric instability associated with the vector-invariant form of the momentum equations due to Arakawa and Lamb (1981). They also provide a modified formulation for the kinetic energy
divergence term which prevents the instability arising. This formulation is available in NEMO from version 3.6 and has been selected in the GO6 configuration. Ducousso et al. (submitted 2017) tested the new formulation in ORCA025 and found





impacts in the most active regions of the model. In particular they found that the new formulation produced a more realistic representation of the equatorial current system in the east Pacific with a stronger and better-defined equatorial undercurrent and increased eddy kinetic energy.

Sensitivity experiments with the GO6 configuration have shown results that are consistent with Ducousso et al. Figure 1 shows the impact of turning on the Hollingsworth et al. formulation on the time-mean vertical diffusivity from the TKE scheme. Particularly noticeable is a reduction in the vertical mixing in the thermocline between about 100m and 300m in the eastern Pacific. (There is also a reduction in the subsurface mixing in the equatorial Indian Ocean - not shown). The reduction in mixing results in an increase in the strength of the equatorial undercurrent as shown in Figures 1 (b) and (c), which show cross sections of the zonal currents at $136°$ W. The equatorial undercurrent (EUC) is substantially stronger in the integration with the Hollingsworth et al. formulation and extends deeper to 250 m. Comparison with the in situ cruise data presented in Figure 1 of Wang (2005) shows that the shape and vertical extent of the EUC with the new formulation appears to be a better match to observations, although the jet is now too strong (1.3 m/s as opposed to about 0.9 m/s in the observations).

### 4.3 Tuning of near-surface mixing

At GO5 a number of parameters that control vertical mixing were adjusted, based on the sensitivity studies of Calvert and Siddorn (2013). The most significant of these tunings was to the ad-hoc parametrisation of near-inertial wave breaking of Rodgers et al. (2014). The e-folding length scale $nn\_htau$, associated with this parametrisation can be varied with latitude. In GO5, $nn\_htau$ was reduced at mid- and high-latitudes, which reduced over-deep summertime mixed layer biases and cold SST biases. However, the summertime mixed layers in the Southern Ocean were then too shallow compared to climatology (see Figure 2 of Megann et al. (2014)).

Figure 2 shows the choices for the latitudinal dependence of the $nn\_htau$ parameter at GO1, GO5 and GO6. Also plotted is the zonal-mean minimum monthly mixed layer depth based on the de Boyer Montégut et al. (2004) climatology. The climatology shows that the summertime mixed layers are on average deeper in the Southern Ocean than in northern mid-latitudes. Based on this and the GO5 results, the $nn\_htau$ parameter has been further tuned at GO6 to be deeper in the southern hemisphere. The results of a sensitivity experiment comparing the GO5 and GO6 vertical mixing tunings are shown in Figure 3. The summertime mixed layer depths in the southern hemisphere have deepened compared to GO5 and are now a better match for the de Boyer Montégut et al. (2004) climatology. This reduces a summertime warm bias in the SST (not shown) which then has a major impact on the sea ice (see Figure 4a and b), increasing the total volume of sea ice in the Southern Ocean year-round. The increase in the volume of sea ice is substantial, particularly in the winter, and is partly due to the suppression of open ocean polynyas in the Weddell Sea. The suppression of the polynyas then shallows the winter time mixed layer depths under the ice as is evident in Figure 3d.

As discussed further in Section 8, we regard the use of the Rodgers et al. (2014) scheme as a stopgap solution to the problem of under-estimated mixing processes in the model and plan to move to a more physically-based parametrisation of these processes in future versions of the model.



## 4.4 Reduced isopycnal diffusion

Met Office coupled models based on GO1 and GO5 have a long-standing warm SST bias in the Southern Ocean which has recently been substantially improved (Williams et al., submitted 2017). This has been largely attributed to atmospheric biases, particularly in the representation of clouds. However, as part of ongoing work to reduce this bias, the isopycnal diffusion

parameter in the ocean was also tuned. Isopycnal diffusion by eddies is responsible for moving heat from depth to the ocean surface (Griffies et al., 2015), especially in the Southern Ocean. So a reduction in the isopycnal diffusion parameter might be expected to cool the surface.

Figures 5 and 6 show the results from sensitivity tests in which the isopycnal diffusion parameter has been reduced from the GO5 value of $300 \text{ m}^2/\text{s}$ to the GO6 value of $150 \text{ m}^2/\text{s}$. Tests were performed with the Met Office GC3.1 coupled model

(Williams et al., submitted 2017) and the forced GO6 configuration. As discussed in Section 8, because of the different surface boundary condition and different atmospheric forcing errors, the biases will generally be different in the coupled and forced contexts. The reduction in the isopycnal diffusion parameter cools the surface in the coupled experiment and largely acts to reduce the warm biases in the southern hemisphere. There is a slight exacerbation of northern hemisphere cold biases. The impact on the SST in the forced test is marginal, with a slight warming (Figure 4d) in the subpolar gyre. The impact on the SSS

in the coupled model is mixed with perhaps some correction of a saline bias in the South Atlantic. In the forced model there is a systematic freshening of the surface waters in the Southern Ocean which largely acts to correct a saline bias.

## 4.5 Icebergs and ice shelves

In GO5 the fresh water input from frozen land masses was represented as surface runoff close to the coastlines of Greenland and Antarctica. For Met Office coupled models prior to GC3.0 (eg. Williams et al. (2015)) the fresh water input from Antarctica

was spread over a large part of the Southern Ocean south of the ACC in order to crudely represent iceberg melting. At GO6 and in GC3.1, the processes involved in the input of fresh water to the ocean from frozen land masses are modelled more realistically using a Lagrangian iceberg model and a parametrisation of ice shelf basal melt. Figure 7 shows the annual mean fresh water input to the ocean from Antarctica using the three methods. In the first two cases the distribution of fresh water input is fixed, but with the interactive icebergs model it will vary depending on the winds and currents.

Marsh et al. (2015) tested the the iceberg module in a CORE2-forced ORCA025 integration similar to the one described in this paper. Their control integration put fresh water input as runoff near the coastline, as was done in GO5. They found that one of the main impacts of the icebergs was to suppress sea ice formation near the coast, since less fresh water is put into the ocean there, reducing stratification and warming the surface layers. With less sea ice formation there was less sea ice overall and a widespread salinification of the surface waters in the Southern Ocean due to reduced melting offshore. We find a similar

impact on the sea ice fraction (Figure 8a) and the surface salinity (Figure 8b). Megann et al. (2014) show that compared to a climatology of the HadISST analysis (Rayner et al., 2003), the sea ice extent in GO5 was realistic in the winter but too low in the summer. Therefore a reduction in the sea ice due to the inclusion of the icebergs scheme is likely to reduce the realism of the sea ice simulation. However, the modelling of the fresh water distribution due to the icebergs is more realistic, and as



discussed in Section 5.2 the reduction in sea ice due to the inclusion of the icebergs scheme is more than offset by the increase in sea ice due to the vertical mixing changes.

As discussed in Section 2.4, GO7 has open cavities under the ice shelves, whereas in GO6 the cavities are closed and fresh water is input at the edge of the ice shelves. Mathiot et al. (2017) show that the inclusion of ice shelf basal melt in a model of the Southern Ocean produces circulations on the Antarctic shelves that are absent in simulations where all the fresh water is put in at the surface. They also show that these circulations are similar in models that have the cavities open and models that close the cavities but input the fresh water through depth as in GO6. In particular, the fact that the fresh water is input at depth produces vertical mixing and, on the West Antarctic continental shelf, draws warmer, saltier water from depth to the surface resulting in a reduction in sea ice formation. In Section 7 we look at the impact of opening the cavities.

## 4.6 Developments to the sea ice model

In parallel to the ocean model development from GO5 to GO6, the sea ice model has been developed from the GSI5.0 (Rae et al. (2015)[2]) configuration used with GO5 to the GSI8.1 configuration described by Ridley et al. (2017). In assessing the GO6 ocean we therefore have to also take into account the impact of changes in the sea ice model. The two main developments in GSI8.1 compared to GSI5.0 are the replacement of the simple temperature-dependent albedo with an explicit calculation of the impact of surface melt ponds on albedo using the topographic melt pond scheme of Flocco et al. (2010), and a change to a multi-layer thermodynamics solver instead of zero-layer thermodynamics. In addition the ocean-ice drag coefficient has been nearly doubled from 0.00536 to 0.01 and for the forced integrations the albedo of bare ice in the visible waveband has been changed from 0.78 to 0.8333 to be more consistent with the value used in the coupled model.

Results of sensitivity experiments show that the introduction of melt ponds (Figure 4 c,d) and multi-layer thermodynamics (Figure 4 e,f) have relatively small impacts on the total sea ice volume. The melt pond scheme slightly reduces the volume of summertime sea ice in the Arctic probably because it reduces the summertime ice albedo, increasing the melt rate for a given insolation. The multi-layer thermodynamics counteracts this tendency by slightly increasing the volume of summertime sea ice in the Arctic. The multi-layer thermodynamics has nonzero thermal inertia in contrast to zero-layer thermodynamics, which increases the time taken for the ice to melt in the summer. The multi-layer thermodynamics also reduces the thickness and extent of wintertime ice in the Antarctic - a slight degradation. The base conductivity in the multi-layer scheme is lower than for the zero-layer scheme so the ocean loses less heat through the ice in the winter, reducing basal ice growth. In the southern hemisphere the changes in the sea ice volume due to the vertical mixing tuning (already discussed - Figure 4 a,b) are much larger than changes in the sea ice volume due to the changes to the sea ice model physics.

---

[2]Rae et al describe the GSI6.0 configuration which only differs from GSI5.0 in the choice of the value of the snow albedo.



## 5 Evaluation of GO6 at $1/4°$ resolution

### 5.1 Model assessment and comparison to GO5

In this section, we evaluate the GO6-GSI8.1 $1/4°$ configuration compared to GO5-GSI5.0, based on a 30-year CORE2-forced integrations. Time averaged fields are taken from the third decade of the 30-year integrations.

Looking first at the temperature, the GO5 model is generally too warm in the Southern Ocean, in the northern subpolar gyres and in the tropics, with cold biases in the subtropical gyres (Figure 9 a,b). The changes at GO6 tend to cool the Southern Ocean south of the ACC and warm the ocean north of the ACC. There is a subsurface cooling in the tropics and a warming of the subpolar gyre in the North Atlantic (Figure 9 c,d). With the exception of the North Atlantic subpolar gyre, the differences generally improve the model (Figure 9 e,f - compare a,b). These near-surface biases are visible in the zonal mean cross section

of temperature (Figure 10). There are also deeper cold biases in the tropics and subtropics. The GO6 changes cool the near-surface ocean south of the ACC and warm it north of the ACC, reducing corresponding cold and warm biases in the top 100m. There is a warming of the Southern Ocean below 200m which introduces a slight warm bias in GO6. The tropical ocean is cooled in the near surface layers, which reduces the near-surface warm bias but to some extent also exacerbates the deeper cool bias.

Regarding surface salinity biases, the GO5 model tends to be too salty in the Southern Ocean, in the subpolar gyres and on the continental shelf in the Arctic, and too fresh in the central Arctic Ocean and in the subtropical gyres (Figure 11). The main impact of the GO6 changes is at high latitudes. The Southern Ocean tends to freshen south of the ACC which reduces the large-scale saline bias. The fresh bias in the central Arctic Ocean tends to be exacerbated.

    Summertime mixed layer depths tend to be too shallow in the Southern Ocean in GO5 and this bias tends to be corrected at

GO6 (Figure 12 a,c,e). In the wintertime GO5 has too-deep mixing in the Atlantic subpolar gyre and the Greenland-Iceland-Norway Seas and also in the Southern Ocean west of Drake Passage and in the Weddell Sea. The very deep winter mixed layers in the Southern Ocean are much reduced at GO6 (Figure 12 b,d,f). The deep mixing in GO5 in the Weddell Sea is related to the presence of a large open-ocean polynya which will be discussed in Section 5.2.

    Figure 13 shows the mean seasonal cycle of integrated sea ice extent and volume in the northern and southern hemispheres.

In the Arctic, GO5-GSI5 and GO6-GSI8.1 both do a reasonable job of simulating the total wintertime sea ice extent compared to a climatology of the HadISST analysis (Rayner et al., 2003), but both models have too small extent in the summer. The total volume is underestimated compared to the Pan-Arctic Ice-Ocean Modeling and Assimilation System (PIOMAS, Zhang and Rothrock (2003)) reanalysis year-round, but especially in the summer. By this metric there is little difference between the two model versions in the Arctic, apart from a slightly faster springtime melting in GO6-GSI8.1 compared to GO5-GSI5.0 and a

marginal increase in the total volume in the autumn and winter in GO6-GSI8.1 compared to GO5-GSI5. In the Antarctic, the wintertime sea ice extent is reasonable compared to HadISST but there is a large underestimation of the summertime extent in both configurations. The wintertime total volume is greatly increased in GO6-GSI8.1 which is partly owing to the fact that the open-ocean Weddell Sea polynyas seen in the GO5-GSI5.0 simulations do not appear in the GO6-GSI8.1 simulations.





To look in more detail at the spatial differences between the GO5-GSI5.0 and GO6-GSI8.1 runs, mean seasonal sea ice concentration fields (Figure 14) and sea ice thickness fields (Figure 15) have been analysed. The analysis shows that GO6-GSI8.1 has improved simulations of sea ice concentration in winter in both the Arctic and the Antarctic. In the Arctic, the concentration is reduced in the Greenland and Barents Seas, mitigating positive biases against HadISST. In the Antarctic, there

is a general increase in sea ice concentration everywhere, most markedly in the Bellingshausen-Amundsen and Ross Seas and in the central Weddell Sea. In these areas, biases against HadISST have been reduced, but the bias is increased west and north of the Antarctic Peninsula (Figure 14 a-e and f-j). In the summer, sea ice concentration over most of the Arctic has slightly reduced in GO6-GSI8.1, a degradation compared to HadISST. In the Antarctic, summertime sea ice concentration tends to increase in GO6-GSI8.1 closer to the coast but reduce further offshore. For the most part, this reduces biases compared to HadISST, except

for the eastern Weddell Sea and the Pacific sector of the Southern Ocean, where the bias has increased (Figure 14 p-t).

Figure 15 shows spatial sea ice thickness distribution in GO6 and difference between GO6 and GO5. There is a moderate increase in the Arctic ice thickness in the winter and moderate decrease in the summer in the GO6 (Figure 14 b,d). In the Antarctic, GO6 has large thickness increases all year around, with the main increase in thicknesses in the western Weddell and Ross Seas (Figure 14 f,h).

**5.2   Attribution of changes in results to model changes**

In this section, we summarise the sensitivities described in Section 4 and show which model changes are responsible for the main changes between GO5 and GO6.

The simulation of the sea ice is an important aspect of the Southern Ocean, feeding back on the ocean hydrography. As shown in Figures 13, 14 and 15 there is substantially more sea ice in the Southern Ocean in the GO6 simulation compared to GO5.

The concentration and the thickness are increased in most areas, especially in the winter. Part of this difference arises in the Weddell Sea where, in the GO5 simulations, large open-ocean polynyas tended to open up in the winter (Megann et al., 2014). At GO6 the polynyas tend to be suppressed. Sensitivity experiments show that the change likely to be mainly responsible for the increased sea ice is the vertical mixing change described in Section 4.3. This deepens the summertime mixed layers in the Southern Ocean and reduces the surface warm bias (see Figures 12 and 9), allowing more sea ice to persist through the summer

and maintaining the stratification which in turn is favourable for sea ice formation in the following winter. Most of the other GO6 changes tend to suppress sea ice formation in the Southern Ocean. The Lagrangian icebergs scheme moves fresh water offshore resulting in saltier and less stable near-shore water. The ice shelf melting parametrisation puts fresh water into the ocean at depth, destabilising the water column and helping to bring warmer, saltier water to the surface. Both of these changes will therefore tend to suppress sea ice formation near the coast and the ice shelf edges and have been shown to result in less

sea ice in sensitivity studies (Marsh et al. (2015), Mathiot et al. (2017)). The developments to the sea ice model have a small impact on the integrated sea ice area and volume in the Southern Ocean (Figure 4); the use of multi-layer thermodynamics tends to slightly reduce the volume of Southern Ocean ice in the winter. It is therefore apparent that in this aspect the vertical mixing change dominates the other changes and results in an improved sea ice simulation. This result is consistent with the





results presented by Heuzé et al. (2015) and Kjellsson et al. (2015) who also show that increased near-surface vertical mixing in the Southern Ocean tends to close open-ocean Weddell Sea polynyas.

The increased Southern Ocean sea ice at GO6 is driven by reduced melting in the open ocean in response to colder summer-time SSTs. The reduced sea ice melt should in turn result in a salinification of the near-surface layers in the Southern Ocean due to reduced fresh water input. However, Figure 11 shows that going from GO5 to GO6 there is a net freshening of near surface waters in the Southern Ocean, which extends to about 150m depth (not shown). This corrects a saline bias compared to EN4. One possible mechanism for this could be the introduction of Lagrangian icebergs which move fresh water offshore, but as discussed in Section 4.5 the net effect of the icebergs is actually to indirectly salinify the surface layers by reducing the production of sea ice and hence reducing the total overall sea ice (Figure 8a). The actual cause of the freshening at GO6 appears to be the reduction in the isopycnal diffusion coefficient (Figure 6c,d), which reduces the amount of warm, salty water upwelled from depth in the Southern Ocean and counteracts the salinification due to reduced sea ice melting.

Figures 9 and 10 show that the main change in the tropics going from GO5 to GO6 is a subsurface cooling at around 100-200m in the Indian and eastern Pacific Oceans. This mitigates a subsurface warm bias visible at about 100m in the GO5 results, but also in the Pacific slightly exacerbates a deeper cold bias. Sensitivity experiments (not shown) suggest that the primary change responsible for this cooling is the use of the Hollingsworth et al. (1983) formulation of the momentum advection scheme. As described in Section 4.2, the subsurface mixing in the central Indian Ocean and eastern Pacific ocean is reduced when the Hollingsworth et al formulation is used. It seems likely that this is causing less heat to be mixed down, giving a cooling of the subsurface layers, which for the most part reduces existing subsurface biases in the tropics. One might expect that if less heat were being mixed down then the SST would show a warming signal. However, in these forced experiments the SST is quite strongly constrained by the forcing and the ocean "sees" an atmosphere with effectively infinite heat capacity.

## 6 Cross-resolution evaluation

In this section we present selected results from the hierarchy of GO6 models.

The large-scale SST biases and SSS biases (Figure 16) are strikingly similar in most respects for the 1°, 1/4° and 1/12° resolutions. In a forced model the near-surface fields are strongly constrained by the atmospheric forcing fields and so the similarity is perhaps unsurprising. The exceptions are in the active regions: the western boundary currents and the ACC, where there are significant differences in the ocean currents, resulting in different heat and salt distributions. The most notable example is in the region of the Gulf Stream Extension and the Grand Banks. In the ORCA1 model, the Gulf Stream Extension fails to be deflected northwards around the Grand Banks and continues in a more zonal path resulting in a strong cold and fresh bias due to the lack of advection of warm, salty water from the south. This is a well-known issue in non-eddy-permitting models (eg. Zhang and Vallis (2007)). By contrast the steering of the Gulf Stream Extension is improved in ORCA025 and ORCA12 and these models tend to have warm, salty biases in this region. Other regions where there are significant differences include the Brazil-Falklands Confluence region and Zapiola Gyre, and the Kuroshio Extension.





The spin up of the Atlantic meridional overturning circulation (AMOC) is shown in Figure 17, together with the mean AMOC for the third decade of the integration. In the three decades of integration there is a marked difference in the behaviour between ORCA1 and the two higher-resolution models. The AMOC at 26N in ORCA1 is between 16 and 18 Sv for the first 15 years and then peaks at 20 Sv in the 1990s. In ORCA025 and ORCA12 the AMOC at 26N increases rapidly over the

first two decades to a peak of 26 Sv in ORCA025 and 30 Sv in ORCA12 and subsequently decreases in both models. For comparison, observation based estimates of the AMOC during the period between April 2004 and October 2012 yield a time mean value of 17.2 Sv with an estimated annual mean r.m.s. uncertainty of 0.9 Sv (McCarthy et al., 2015). The ORCA025 and ORCA12 peak values are well outside the range of the RAPID observations, and are still too large at the end of the integration. This behaviour in the initial spin up period is specific to forced integrations; the spin up of a coupled model using a similar

ORCA025 configuration and similar initial conditions does not show this large peak (Graham, 2017).

ORCA025 and ORCA12 exhibit very deep wintertime mixed layer depths and a gradual salinification of the water masses in the central Labrador Sea over this period (not shown), neither of which is present to the same degree in the ORCA1 integration. Danabasoglu et al. (2014) show that the strength of the AMOC is well correlated with the depth of the wintertime mixed layers in the Labrador Sea across a range of models forced with the CORE2 forcing set. The models in their study which have the

deepest mixed layers also have a salty bias in the Labrador Sea, which may be the cause of the deep mixed layers. It seems likely that the large AMOC values in the higher resolution models in the present study are linked to the salinification and deep wintertime mixing in the Labrador Sea. Treguier et al. (2005) describe the salinification of the Labrador Sea in the early spin up of a range of forced models and ascribe this to erroneous salt transports by the ocean currents, either in the main subpolar gyre or in the East Greenland Coastal Current.

In Figure 18 we show globally integrated temperature and salinity drifts from initial conditions for the three models calculated from annual-mean fields. Because the models are initialised and evaluated against the same temperature and salinity climatologies, this is another way of looking at the evolution of the model biases against climatology. The temperature drifts across the three models all show a warming in the top 200m and a cooling in deeper waters centred at about 400m. The nearsurface warming appears to equilibrate quickly but the subsurface cooling in ORCA025 and ORCA12 is still drifting after 30

years of integration. The cooling is much smaller in ORCA1 and it is not clear that it is showing an ongoing drift. For the salinity all three models show a freshening centred at about 200m which is not equilibrated after 30 years of integration.

The subsurface cooling and freshening trends in the global mean are dominated by cooling and freshening trends in the tropical and subtropical Atlantic and Pacific (not shown). All three models show a similar cooling and freshening centred at about 200m in the tropical Pacific. The differences in the global mean between the three models are largely due to different

behaviour in the tropical and subtropical Atlantic, where the ORCA1 model shows a moderate cooling centred at 500m, but the ORCA025 and ORCA12 show a substantial cooling and freshening of the top 1000m, which is more marked in ORCA025. It is possible that this is linked to the different AMOC behaviour between the models in this period, with a much more intense AMOC in ORCA025 and ORCA12 resulting in colder and fresher water being advected from the south. The shortness of the integrations means we cannot draw conclusions about long-term drifts. Nevertheless the substantial differences in the drifts in



this initial spin up period at different resolutions demonstrate the potential of differing ocean model responses to the same flux forcing to result in very different heat and salt distributions.

## 7  GO7: Opening the ice shelf cavities

The single difference between GO6 and GO7 is how the ice shelf melting is distributed. In GO6, the ice shelf melting is spread in depth along the ice shelf front (similar to simulation R_PAR in Mathiot et al. (2017)). In GO7, the ice shelf cavities are opened and the ice shelf melting is prescribed at the ice shelf/ocean interface (similar to simulation R_ISF in Mathiot et al. (2017)). The total fresh water from ice shelf melt is the same in GO6 and GO7 for each ice shelf. The melt pattern used in GO7 is the one described in Mathiot et al. (2017). The bathymetry and the ice shelf draft beneath the ice shelf comes from BEDMAP2 (Fretwell et al., 2013) as in Mathiot et al. (2017). As for the GO6 results, the results discussed here are based on the climatology of the last 10 years of the integrations (1996-2005).

In the open ocean as well as on most of the East and West Antarctica continental shelves, the temperature and salinity properties are similar in both GO6 and GO7 simulations (Figure 19 b,d). On the Filchner and Ronne continental shelf, the ocean circulation over the shelf is very different in GO7 to the one simulated in GO6 because of the new pathway beneath the ice shelves. The circulation of High Salinity Shelf Water beneath the ice shelf leads to a different salinity distribution over the Filchner shelf. The salinity gradient between the east and west side is decreased and improved. The shelf waters are saltier on the east side due to the outflow of Ice Shelf Water from Ronne Ice Shelf cavity, but are fresher on the west side. However, the overall the Filchner shelf is still too salty.

The opening of ice shelf cavities also leads to some modification in the intrusion of Circumpolar Deep Water (CDW) over the Ross and Bellingshausen shelves. Over the Ross continental shelf, the intrusion of CDW onto the east side of the Ross continental shelf is weaker in GO7 and the shelf is colder. The intrusion of CDW is not realistic (Figure 19a). In the Bellingshausen Sea, the opening of the ice shelf cavities leads to a slight warming of the continental shelf (Figure 19 b,d). However, the West Antarctic shelf is still too cold and too fresh compared to the observations (Figure 19a).

## 8  Summary and future plans

GO6 and GO7 are the latest versions of the UK Global Ocean configuration, developed as a collaboration between the Met Office, the National Oceanography Centre and the British Antarctic Survey. They will be used in a variety of applications, notably in coupled mode in the UK contributions to CMIP6, and in associated OMIP simulations. GO6 and GO7 are developments of the GO5 configuration described by Megann et al. (2014), with updates to the core dynamics, some tuning of mixing coefficients and an improved representation of the cryosphere, with the capability of representing circulation and melting under ice shelves and the introduction of interactive icebergs. Whereas GO5 was defined for just one horizontal grid at a nominal $1/4°$ resolution, GO6 is a largely traceable hierarchy of three horizontal resolutions at nominal $1°$, $1/4°$ and $1/12°$ resolutions. GO7 currently only exists at $1/4°$ resolution.





We have presented results from 30-year integrations forced with the CORE2 data set. Comparing the GO6 and GO5 configurations at $1/4°$ resolution, GO6 has deeper summertime mixed layers in the Southern Ocean and cooler and fresher surface waters south of the ACC. These changes reduce mixed-layer depth biases against the de Boyer Montégut et al. (2004) data set and reduce near-surface temperature and salinity biases measured against the ESA CCI and EN4 datasets. The simulation

of sea ice in the Southern Ocean is also improved with greater extent of summertime sea ice (which is nevertheless still too small compared to observations), and year-round thicker sea ice. The improvements in the mixed layer depth, the sea ice, and the temperature and salinity fields in the Southern Ocean are driven by the tuning of the near-surface vertical mixing and the adjustment of the coefficient of isopycnal mixing. In the subsurface below 100m there is a slight exacerbation of temperature biases in GO6 compared to GO5, with warm biases in the Southern Ocean and increased cold biases in the tropics.

Selected results from forced integrations with the hierarchy of resolutions at GO6 have been presented. The different resolutions span the transition from non-eddy-resolving to (partially) eddy resolving and show quite different near-surface horizontal circulations, resulting in large differences in the temperature and salinity fields in active regions such as the western boundary currents. There are large differences in the initial adjustment of the AMOC which are possibly linked to differences in the deep drifts in the Atlantic.

The opening of the ice shelf cavities in GO7 leads to localised impacts on the temperature and salinity fields. There is a better distribution of the High Salinity Shelf Water over the Filchner continental shelf and the opening of the cavities tends to decrease the extension of the Circumpolar Deep Water over Ross Sea and tends to increase it over Bellingshausen Sea.

This paper has described results from integrations forced with the CORE2 atmospheric forcing data set (Large and Yeager, 2009). As described in Griffies et al. (2009) two major sources of error in forced integrations are errors in the atmospheric

forcing fields and errors associated with the use of mixed boundary conditions, whereby the sea surface temperature field is constrained more strongly than the sea surface salinity field. Because of the different atmospheric forcing errors, the same ocean model configuration will generally exhibit different biases in the context of a coupled model than in the context of a forced integration (as seen for example in Section 4.4). The forced configuration described here is used for initial development and exploration of sensitivities since it is less resource-intensive than the coupled model. The resulting prototype configuration

is then incorporated in the coupled model where further tuning of ocean parameters may take place. The choice of the isopycnal diffusion coefficient discussed in Section 4.4 is an example of a parameter tuning driven by the coupled model results. Since GO6/GO7 is intended for use in both coupled and forced configurations, any tuning is a necessary compromise, but we have generally prioritised the coupled model and accepted that the tunings may be less than optimal for the forced model. The results presented in this paper are from the final configuration as used in the coupled model. An example of where this is

slightly non-optimal in the forced context is the thickness of the Arctic sea ice which compares well with observations in the coupled GC3.1 model (Ridley et al., 2017), but is too thin in the forced model, especially in the summer (Figure 13).

The next round of development (GO8) will focus on near-surface vertical mixing, overflows in the North Atlantic, and spurious numerical mixing.

The vertical mixing closure in NEMO under-represents several processes, including Langmuir turbulence (Grant and Belcher,

2009; Belcher et al., 2012) and mixing due to shear spiking at the mixed layer base (Large and Crawford, 1995; Skyllingstad

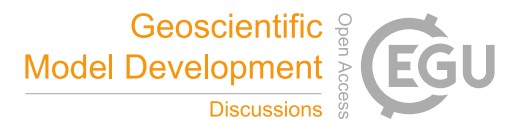



et al., 2000; Grant and Belcher, 2011). These two processes in particular are thought to be important for near-surface mixing globally (Belcher et al., 2012) and so the NEMO vertical mixing closure is underestimating important sources of mixing. This manifests in particular as too shallow summertime mixed layers in the Southern Ocean. The Rodgers et al. (2014) scheme is an ad hoc attempt to compensate for this missing mixing. The UK OSMOSIS project has attempted to characterise near-surface

oceanic mixing more accurately through an observational campaign and a new mixing scheme based on the results of large eddy simulations, which includes the effect of Langmuir turbulence. We plan to incorporate this more physically-based scheme at GO8.

The poor representation of dense overflow currents is a long-standing issue in $z$-coordinate models, in which the stepwise bathymetry causes density currents to mix too much with the ambient water masses. The problem is known to reduce as the

horizontal and vertical resolution increase. Winton et al. (1998) suggest that to adequately resolve the Denmark Strait overflow a horizontal resolution of 3-5km and a vertical resolution of 30-50m are required. This is significantly higher resolution than the global models described in this paper. In developing GO8 we intend to experiment with embedding two-way nests covering the important overflows as well as to explore alternative vertical coordinates, such as $s$-coordinate which better represents the density currents.

There are more general issues with spurious numerical mixing in $z$-coordinate models, which tend to undermine the models' ability to preserve water masses over time (Griffies et al. (2000), Ilıcak et al. (2012), Megann (2017)). Another focus of the development in GO8 is to explore ways of reducing this mixing, with one option being the $\tilde{z}$ formulation of vertical levels due to Leclair and Madec (2011).

## 9   User Manuals

The user manuals for the NEMO and CICE modelling codes are available online at:

- https://www.nemo-ocean.eu/wp-content/uploads/NEMO_book.pdf

- http://oceans11.lanl.gov/trac/CICE/raw-attachment/wiki/WikiStart/cicedoc.pdf

Namelist settings for the GO6 and GSI8.1 configurations are included in the Supplement.

## 10   Code availability

The ocean model code is available from the NEMO website (www.nemo-ocean.eu) under the CeCILL free software license (http://www.cecill.info/). On registering, individuals can access the Fortran code using the open source subversion software (http://subversion.apache.org/). The base code used for the integrations presented in this paper is in revision 7750 of the following branch:

- branches/UKMO/dev_r5518_GO6_package



This consists of the NEMO v3.6 release with the addition of GO6-specific changes. In addition the following branch at revision 6568 is required for the ORCA12 configuration:

– branches/UKMO/dev_5518_shlat2d

The sea ice model code is freely available from the Met Office Science Repository (https://code.metoffice.gov.uk/trac/cice) under the CICE copyright agreement (http://oceans11.lanl.gov/trac/CICE/wiki/CopyRight). As for the NEMO repository, registration is required and then the Fortran code is available using subversion. The code used for the integrations presented in this paper consisted of a number of branches of the CICE code. These branches have subsequently been merged into a single package branch at revision 235:

– branches/pkg/Config/vn5.1.2_GSI8.1_package_branch

Preprocessing keys required in building GO6-GSI8.1 are listed in the Supplement.

## 11 Data availability

Input data files required to run the simulations described in this paper, and results from the simulations are archived at the Met Office and available for research use through the Centre for Environmental Data Analysis JASMIN platform (http://www.jasmin.ac.uk/); please contact the authors for details.

*Author contributions.* D. Storkey prepared the manuscript with contributions from co-authors. D. Storkey, A. Blaker, P. Mathiot and A. Megann performed and analysed the main assessment integrations. Y. Aksenov, E. Blockley, D. Calvert, T. Graham, H. Hewitt, P. Hyder, T. Kuhlbrodt, J. Rae, and B. Sinha were involved in the development of the GO6 and GSI8.1 configurations, performed sensitivity experiments and assisted with the evaluation of the main integrations.

*Acknowledgements.* Met Office authors acknowledge support by the Joint DECC/Defra Met Office Hadley Centre Climate Programme (GA01101), the Ministry of Defence, the Public Weather Service, and from the Copernicus Marine Environment Monitoring Service. NOC authors would like to acknowledge the support of NERC national capability funding. We acknowledge use of the MONSooN system, a collaborative facility supplied under the Joint Weather and Climate Research Programme, a strategic partnership between the Met Office and the Natural Environment Research Council.



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





**Table 1.** Table giving GO6 parameters and settings that vary between resolutions.

|  | ORCA1 | ORCA025 | ORCA12 |
|---|---|---|---|
| **lateral viscosity** | laplacian $20000m^2/s$ | bilaplacian $-1.5e + 11m^4/s$ | bilaplacian $-1.25e + 10m^4/s$ |
| **isopycnal tracer diffusion** | $1000m^2/s$ | $150m^2/s$ | $125m^2/s$ |
| **timestep** | $2700s$ | $1350s$ | $360s$ |





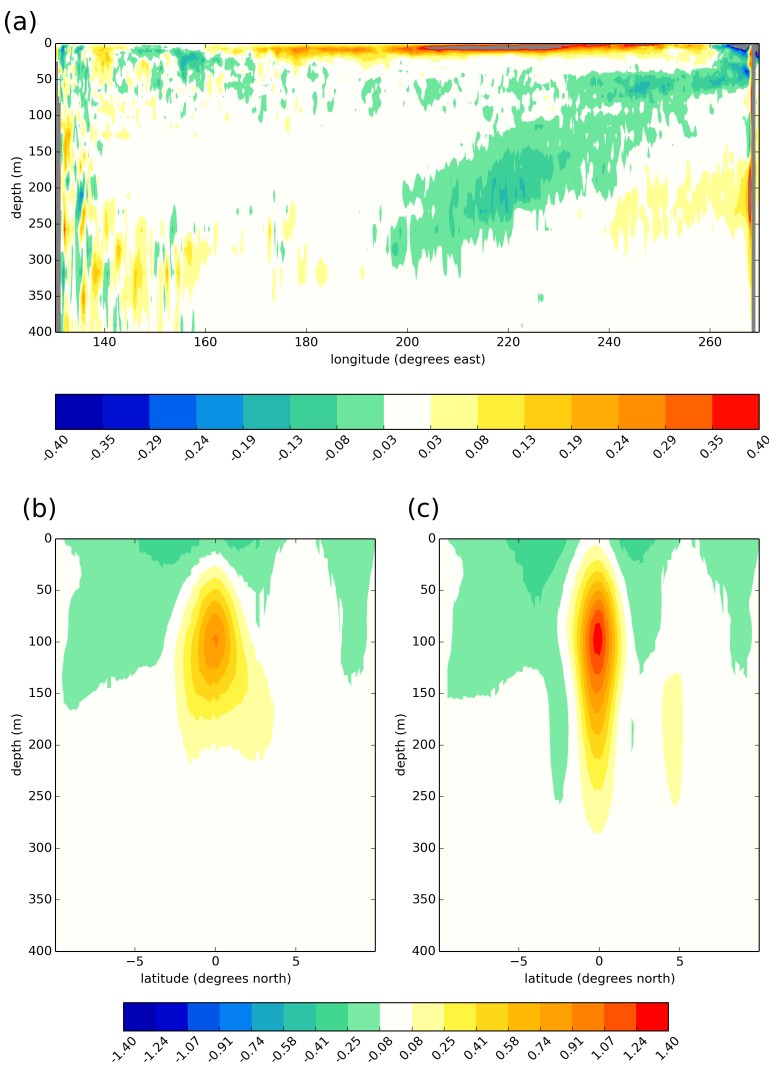

**Figure 1.** Figure showing the impact of turning on the Hollingsworth et al. (1983) scheme for momentum advection: **(a)** cross section along the equator in the Pacific of the difference between experiment and control of the diagnosed vertical tracer diffusivity (m$^2$/s) from the TKE scheme; **(b) - (c)** cross sections of zonal velocity (m/s) at 136W in the Pacific - **(b)** without Hollingsworth; **(c)** with Hollingsworth. Model fields are 5 year means from the second 5 years of a 10-year spin up.



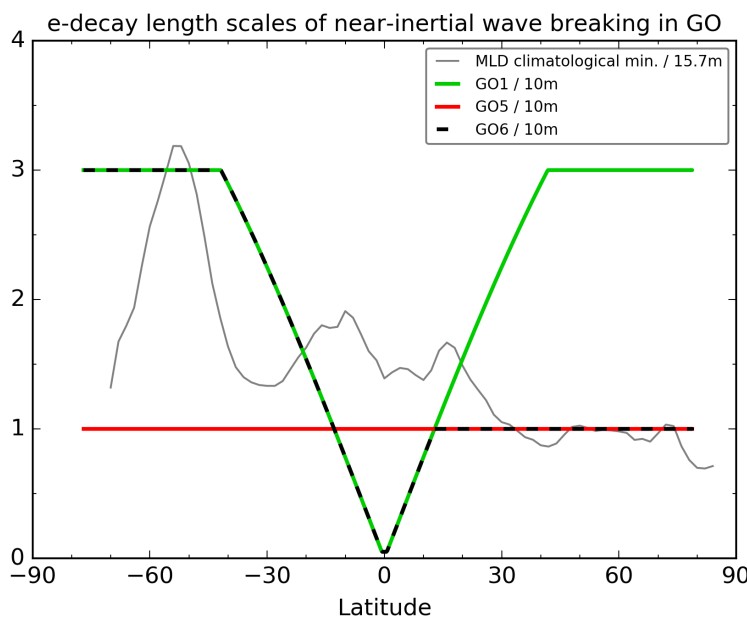

**Figure 2.** Figure showing the variation of the $nn\_htau$ length scale with latitude for different GO configurations. The length scale is normalised by 10 m. Also shown as a grey line is the zonal-mean monthly minimum mixed layer depth from the de Boyer Montégut et al. (2004) climatology. This has been normalised by 15.5 m.





**Figure 3.** Results from a sensitivity study where the $nn\_htau$ length scale was tuned. A mean monthly climatology of the mixed layer depth field in metres was calculated for the third decade of the spin up (1996-2005). The pointwise minimum and maximum monthy values have then been calculated and plotted as anomalies against the same quantities from the de Boyer Montégut et al. (2004) climatology. Hashed regions indicate the presence of sea ice at any time during the year. **(a),(b)** minimum and maximum mixed layer depth anomalies for integration with $nn\_htau$ from GO5; **(e),(f)** as (a),(b) with $nn\_htau$ from GO6; **(c),(d)** the difference fields between the two integrations.

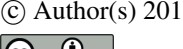


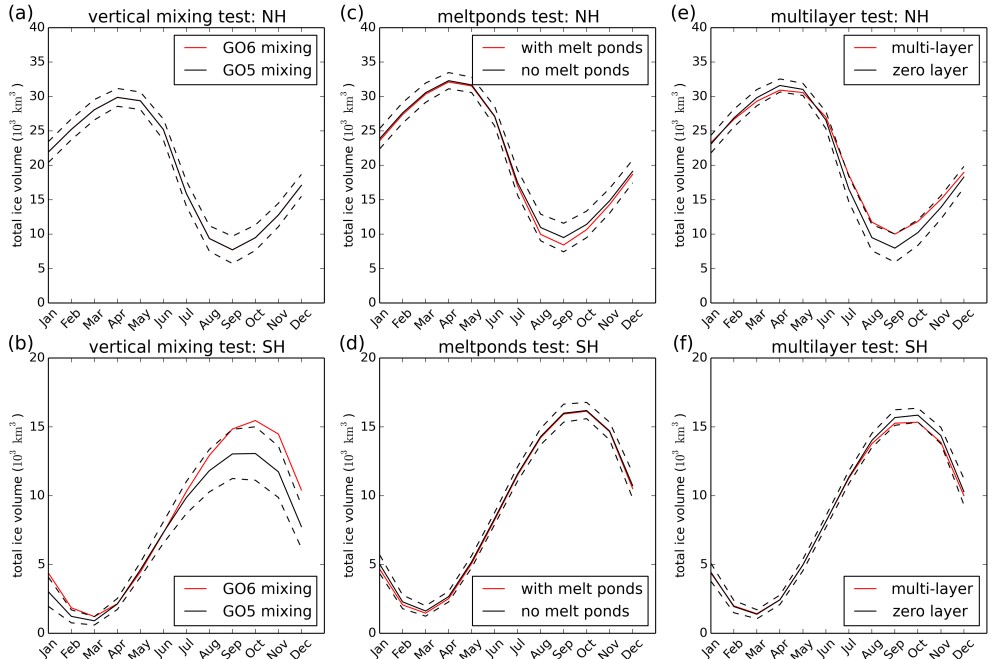

**Figure 4.** Plots showing the mean seasonal cycle from 10-year forced integrations of integrated sea ice volume for northern hemisphere (top row) and southern hemisphere (bottom row). Results are shown from three sensitivity experiments: **(a)** and **(b)** tuning of near-surface vertical mixing; **(c)** and **(d)** introduction of explicit calculation of the impact of melt ponds on albedo versus a simple temperature-dependent albedo; **(e)** and **(f)** introduction of multi-layer thermodynamics versus zero-layer thermodynamics. In all cases the results with the new settings are in red and with the old settings in black. The dashed lines show the standard deviation of the annual mean values for the control integration in each case.





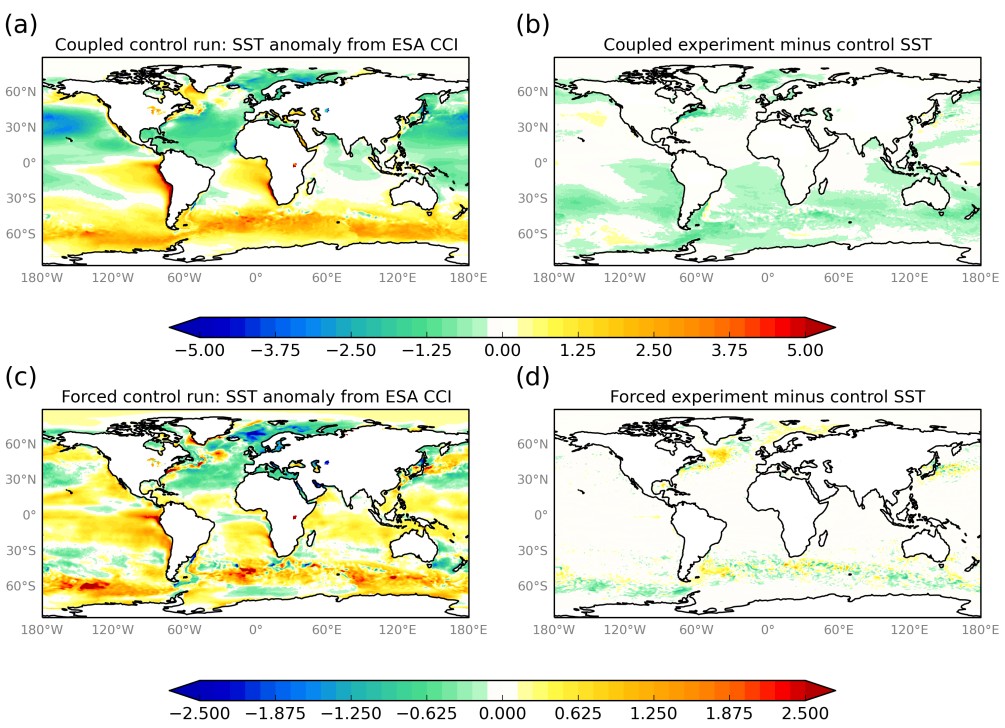

**Figure 5.** Results from sensitivity tests of halving the isopycnal diffusion parameter (from 300 m$^2$/s to 150 m$^2$/s) in: **(a)** and **(b)** the Met Office GC3 coupled configuration (Williams et al., submitted 2017); and **(c)** and **(d)**) the forced GO6 configuration. Plot **(a)** shows 15-year mean SST anomalies (K) against ESA CCI (Merchant et al., 2014) from a spin-up of a coupled model with the original diffusion coefficient and plot **(b)** shows the impact of reducing the isopycnal diffusion parameter. Plots **(c)** and **(d)** show the same fields for the second 5 years of a 10-year CORE2-forced GO6 test.





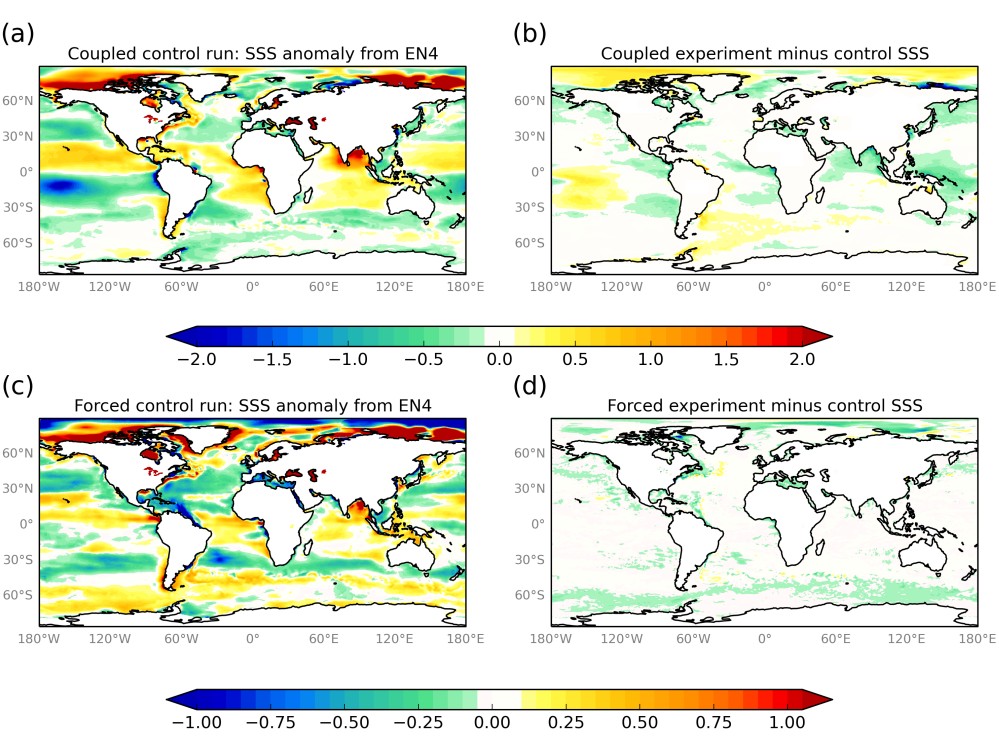

**Figure 6.** As for figure 5 but for sea surface salinity (psu). Anomalies are against the EN4 analysis (Good et al., 2013).



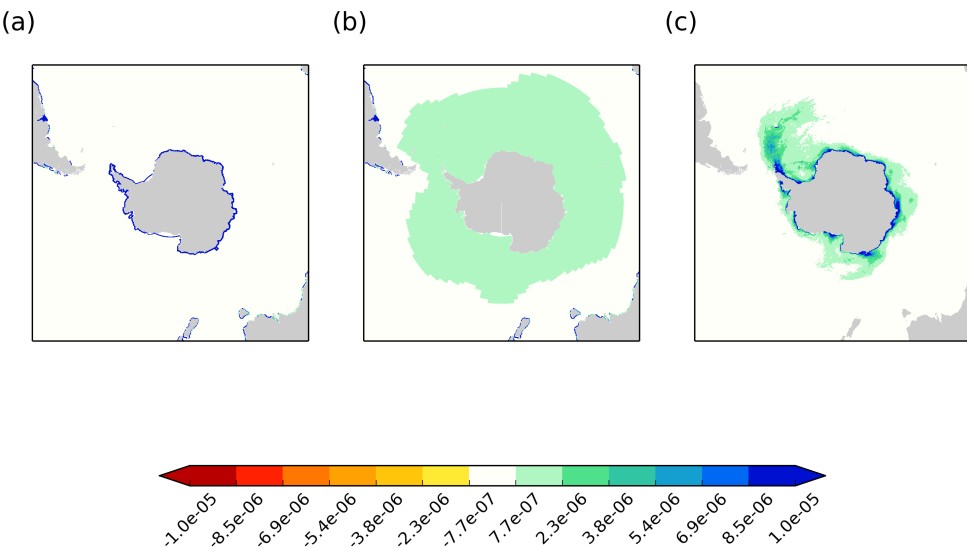

**Figure 7.** Annual mean freshwater input (kg/m$^2$/s) from Antarctic land mass to ocean: **(a)** Annual mean freshwater input where freshwater input from Antarctica is represented as runoff close to the coastline; **(b)** as for (a) but with the freshwater input from Antarctica spread over the Southern Ocean to crudely repesent iceberg melt; **(c)** Annual mean freshwater input from Antarctica using the ice shelf parametrisation and interactive icebergs scheme.





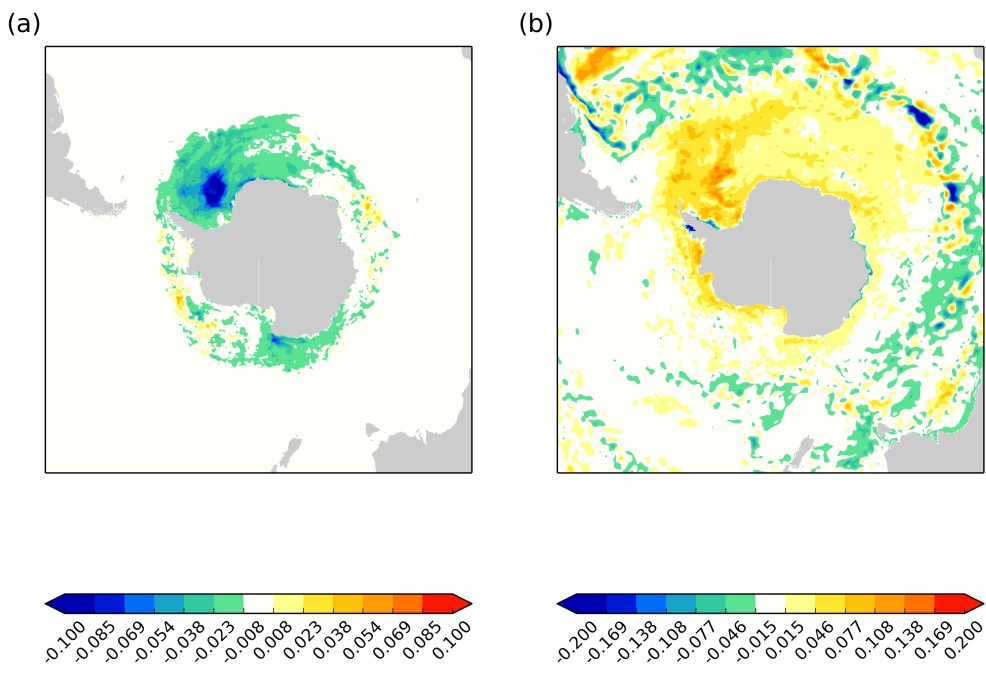

**Figure 8.** Impact of inclusion of active icebergs scheme instead of runoff-type input used in GO5: **(a)** difference in sea ice fraction, icebergs minus no-icebergs; **(b)** difference in surface salinity (psu), icebergs minus no-icebergs. In both cases fields are meaned over the second 5 years of a 10-year spin up.



**Figure 9.** Potential temperature (K): anomalies against climatology and GO6-GO5 differences. **(a)** GO5 SST anomaly against ESA CCI (Merchant et al., 2014); **(b)** GO5 100m potential temperature anomaly against EN4 v1.1 (Good et al., 2013); ; **(c)** Difference GO6 minus GO5 SST; **(d)** Difference GO6 minus GO5 100m potential temperature **(e)** GO6 SST anomaly against ESA CCI; **(f)** GO6 100m potential temperature anomaly against EN4 v1.1. Model fields are time-means over the third decade of the integration.



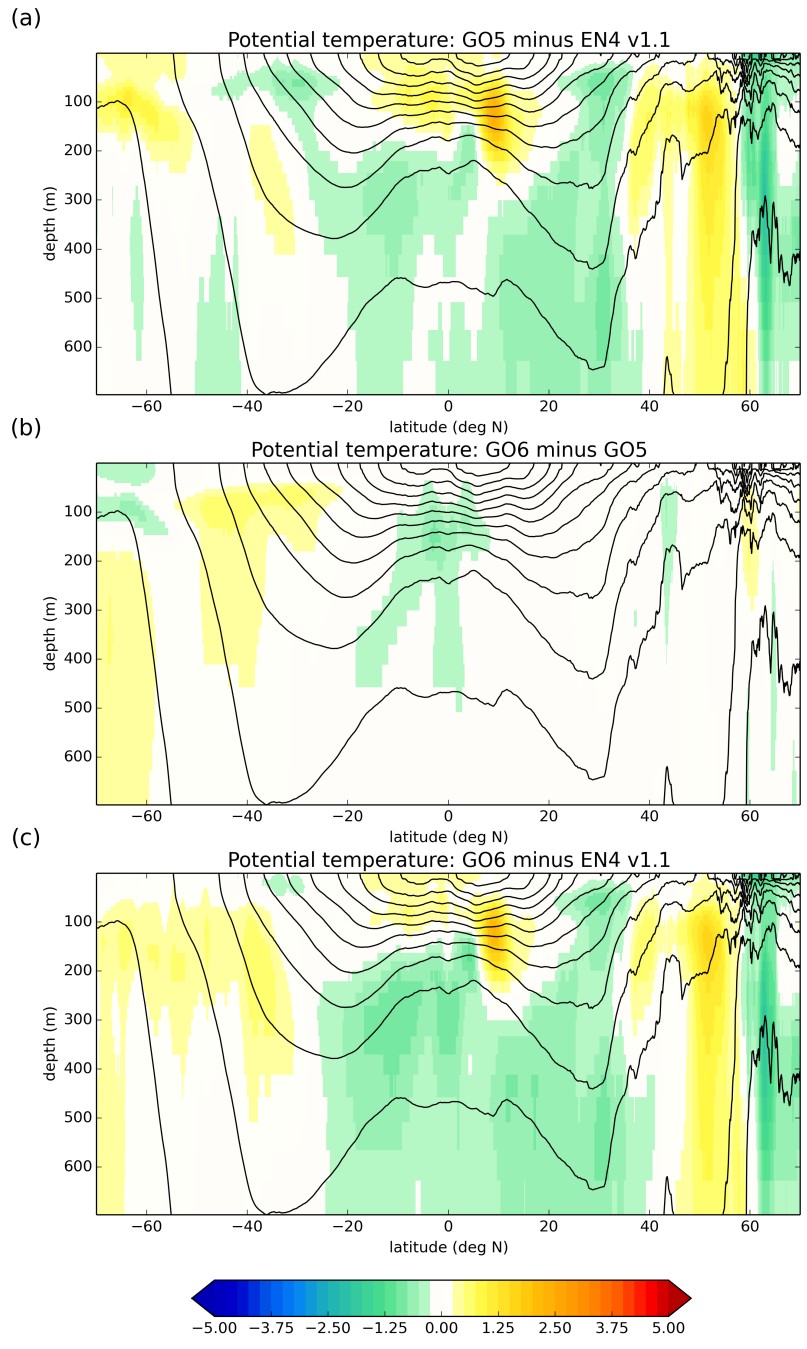

**Figure 10.** Cross sections of zonal mean potential temperature anomalies (K), showing latitude in degrees on the x-axis and depth in metres on the y-axis. **(a)** GO5 anomaly against EN4 v1.1 (Good et al., 2013); **(b)** Difference GO6 minus GO5; **(c)** GO5 anomaly against EN4 v1.1. Contours are zonal mean potential density $\sigma_0$ from the GO6 integration with a contour interval of 0.5 kg/m$^3$. Model fields are time-means over the third decade of the integration.




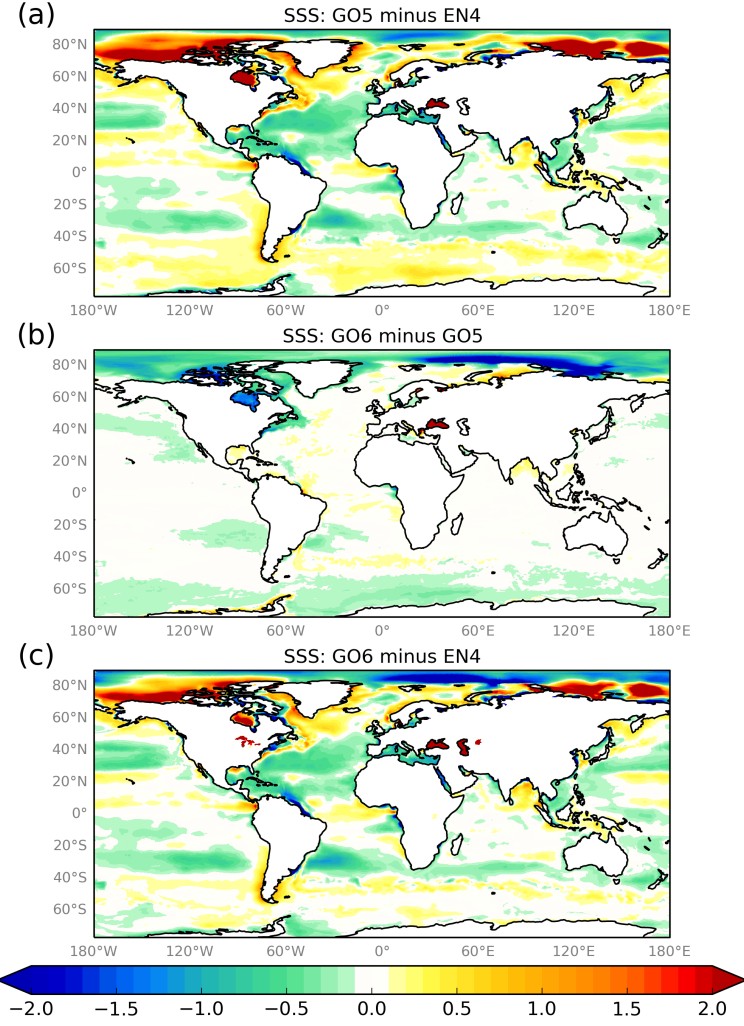

**Figure 11. (a)** GO5 SSS anomaly (psu) against EN4 v1.1 (Good et al., 2013); **(b)** Difference GO6 minus GO5 SSS; **(c)** GO6 SSS anomaly against EN4 v1.1. Fields meaned over the last 10 years of the 30 year spin up.



**Figure 12.** Pointwise minimum and maximum monthly mixed layer depth anomalies (m) and differences in metres calculated as for Figure 3. Hashed regions indicate the presence of sea ice at any time during the year. **(a)** GO5 minimum monthly mixed layer depth against de Boyer Montégut et al. (2004); **(b)** GO5 maximum monthly mixed layer depth against deBoyer Montegut; **(c)** Difference GO6 minus GO5 minimum monthly mixed layer depth; **(d)** Difference GO6 minus GO5 maximum monthly mixed layer depth; **(e)** GO6 minimum monthly mixed layer depth against deBoyer Montegut; **(f)** GO6 maximum monthly mixed layer depth against deBoyer Montegut.



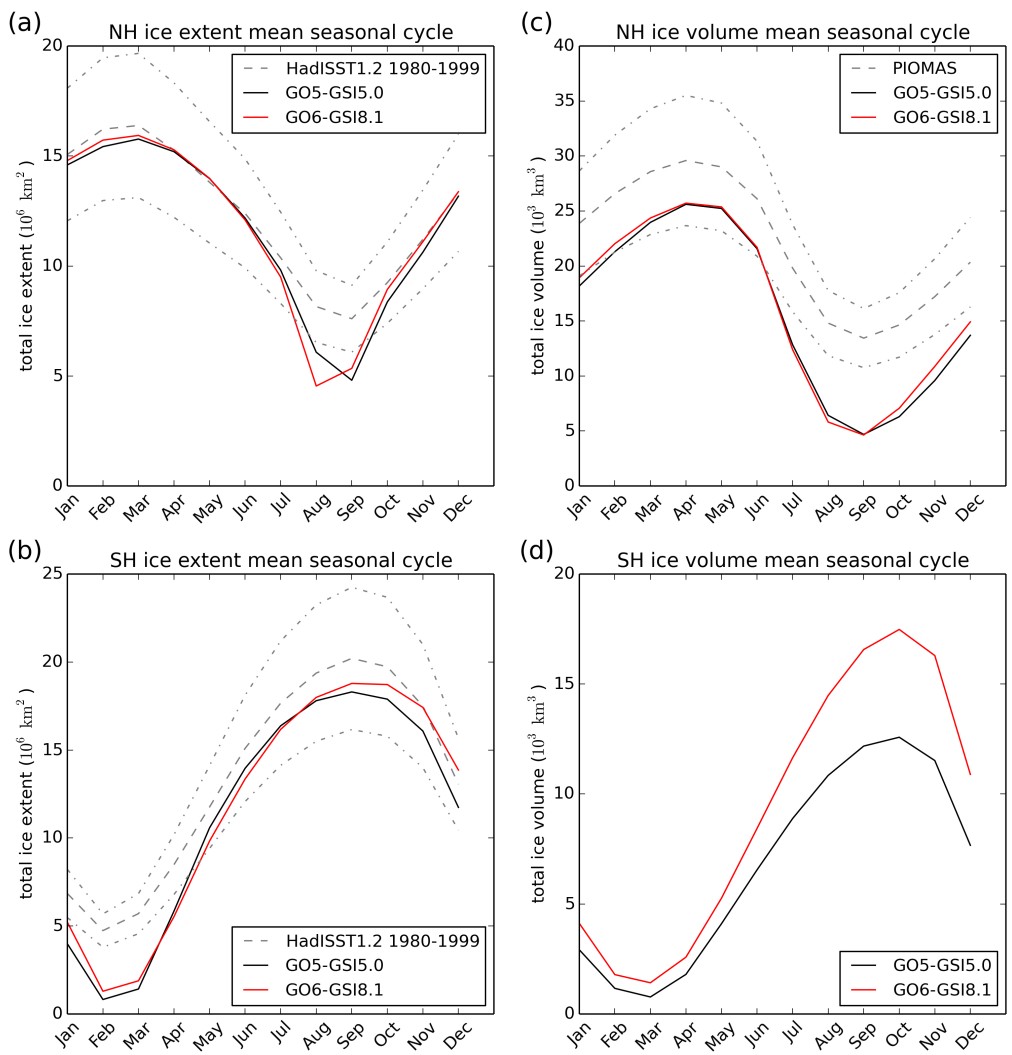

**Figure 13.** Mean seasonal cycles for integrated sea ice extent and volume for the two hemispheres for GO5-GSI5 and GO6-GSI8.1. Sea ice extent is calculated as the integral of the area of grid cells where the sea ice concentration is greater than 15%. The meaning period is 1977-2004. **(a)** Mean seasonal cycle of sea ice extent in the Arctic from GO5-GSI5 (black line) and GO6-GSI8.1 (red line) compared with a climatology of the HadISST analysis (Rayner et al., 2003) (grey lines; dashed lines indicate +-20%); **(b)** Same as (a) for the Antarctic; **(c)** Mean seasonal cycle seaice volume in the Arctic from GO5-GSI5 (black line) and GO6-GSI8.1 (red line) compared with PIOMAS (Zhang and Rothrock, 2003) reanalysis (grey lines; dashed lines indicate +-20%); **(d)** Same as (c) for the Antarctic for GO5-GSI5 and GO6-GSI8.1.



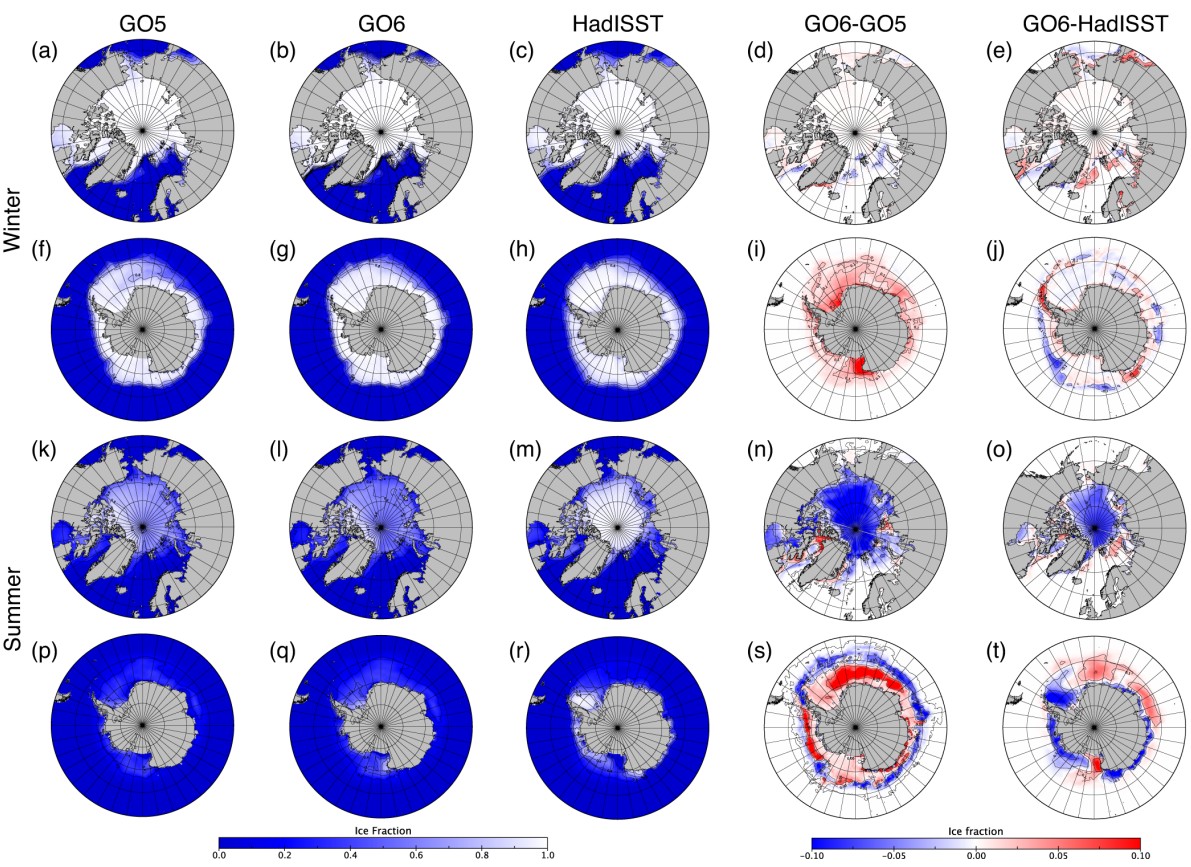

**Figure 14.** Seasonal-mean multi-annual sea ice concentration for summer and winter, averaged for 1978-2005 in GO5-GSI5.0, GO6-GSI8.1, and the HadISST analysis (Rayner et al., 2003). Panels **(a-e)** - show GO5-GSI5.0, GO6-GSI8.1 and HadISST ice concentration and the concentration difference between GO6-GSI8.1 and GO5-GSI5.0 and the bias between GO6-GSI8.1 and HadISST respectively for DJF in the Northern Hemisphere (NH); **(f-j)** are the same as above but for JJA in the Southern Hemisphere (SH); **(k-o)** are the same as (a-e) but for JJA in the NH; **(p-t)** are the same as (f-j) but for DJF in the SH.



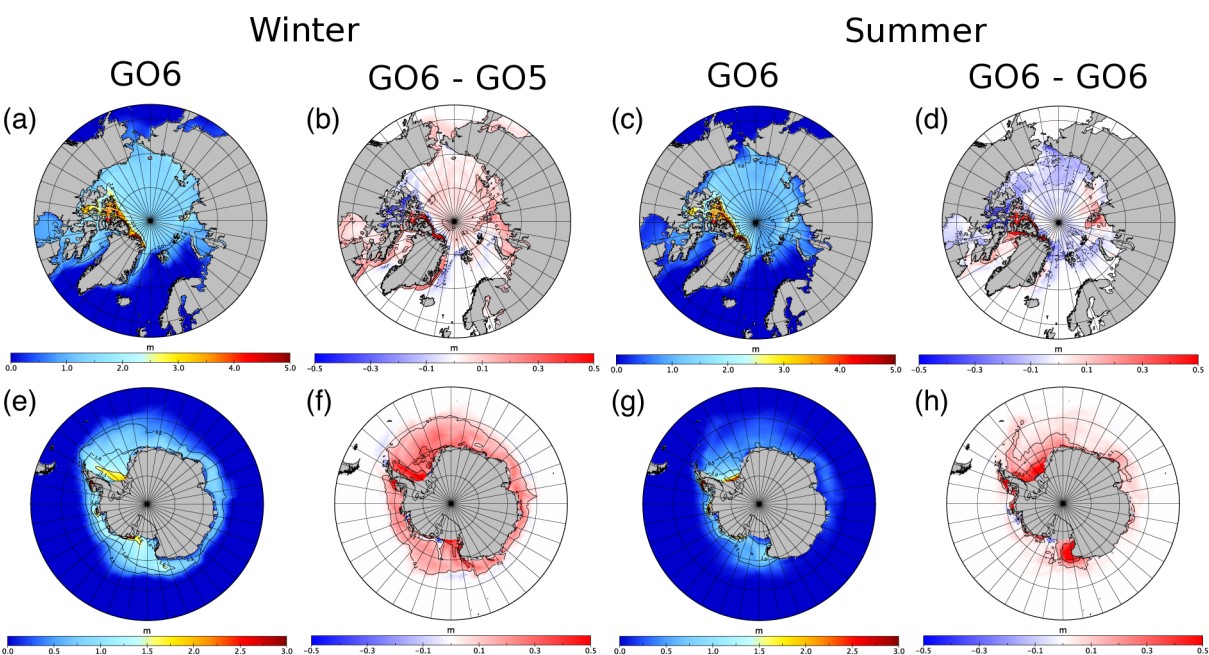

**Figure 15.** Seasonal-mean multi-annual sea ice thickness (m) in the summer and winter, averaged for 1978-2005 in GO6-GSI8.1, GO5-GSI5.0 and the differences between two runs. Panels **(a-d)** - show sea ice thickness in GO6-GSI8.1 and the difference between GO6-GSI8.1 and GO5-GSI5.0 for DJF (a,b) and JJA (c,d) in the Northern Hemisphere (NH); **(e-h)** are the same as above but for JJA (e,f) and (DJF) (g,h) in the Southern Hemisphere (SH).



**Figure 16.** Cross resolution results: Left-hand column: model SST anomalies (K) against ESA CCI (Merchant et al., 2014); right-hand column: model SSS anomalies (psu) against the EN4 v1.1 analysis (Good et al., 2013). In all cases the model fields are 10-year means for the third decade of the spin up (1996-2005).



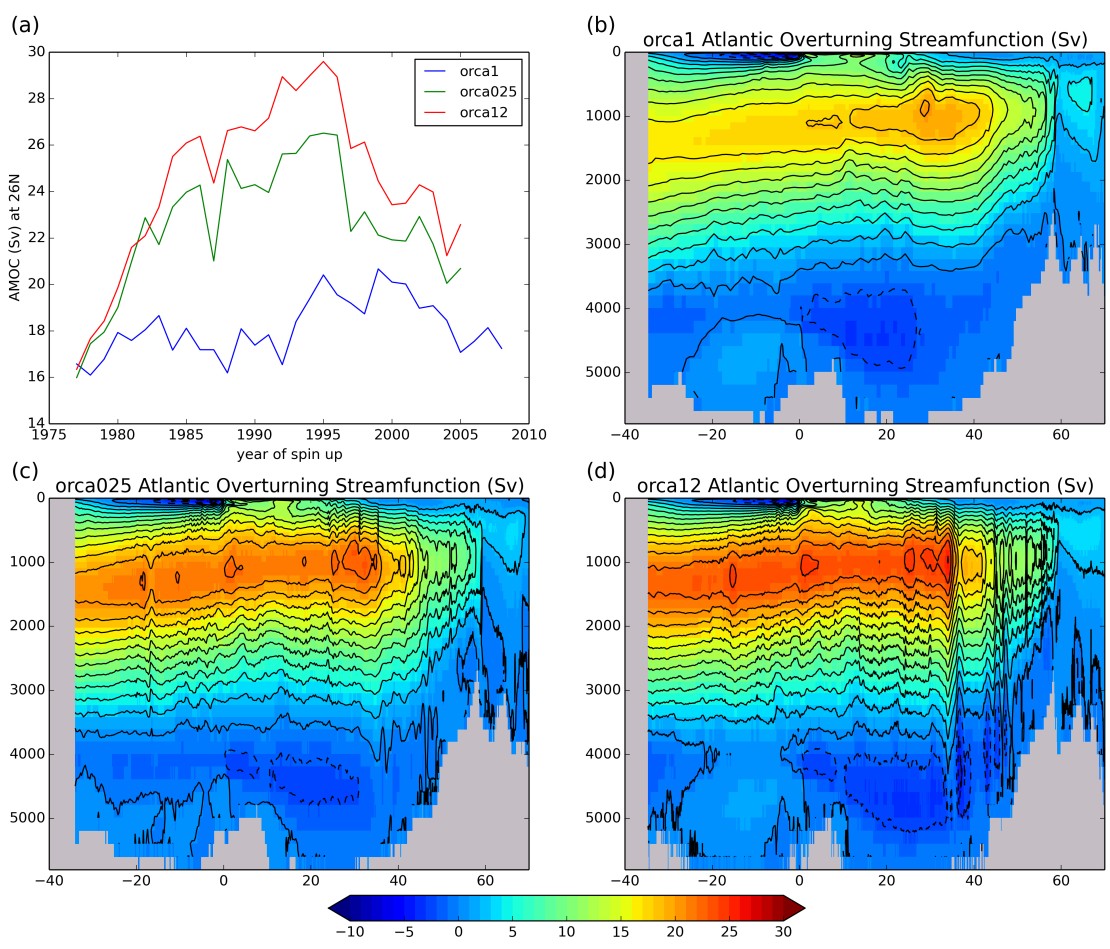

**Figure 17.** Cross resolution results: Atlantic Overturning Streamfunction. **(a)** Timeseries of annual-mean AMOC at 26N for 1976-2005; **(b)-(d)** AMOC (Sv) meaned over the third decade of the spin up (1996-2005).

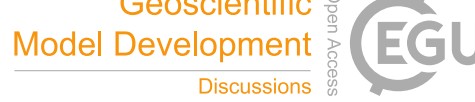

**Figure 18.** Cross resolution results, Left-hand column: global-mean model potential temperature drift from initial conditions (K); right-hand column: global-mean model salinity drift from initial conditions (psu).





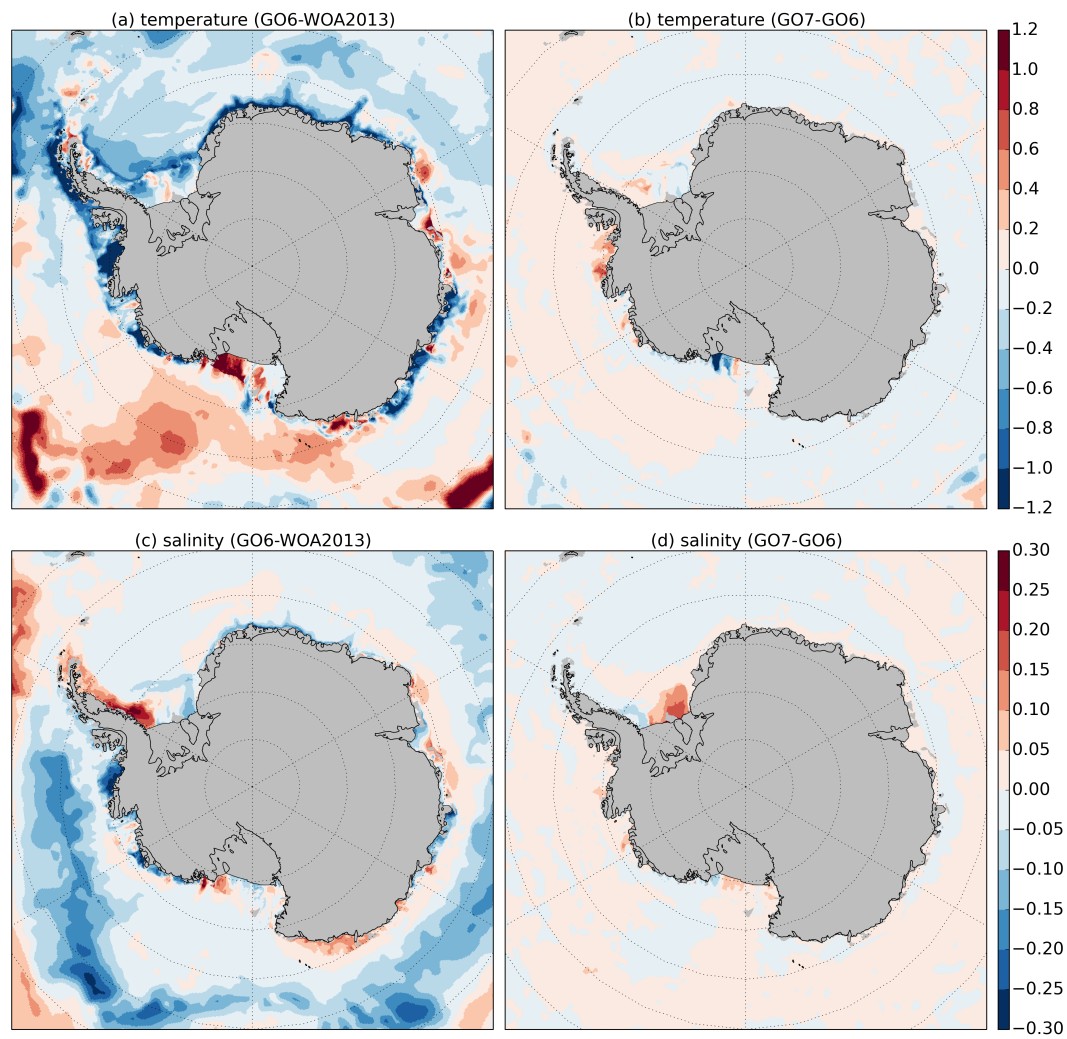

**Figure 19.** Maps of potential temperature (K) and salinity (psu) averaged between 300 and 1000 m. **(a, c)** differences between GO6 and WOA2013 (Locarnini et al. (2013) and Zweng et al. (2013)) data; **(b, d)** differences between GO7 and GO6. Grey shading represents areas covered by ice shelves or ice sheets, or with bathymetry shallower than 300 m.