# Peer review of "UK Global Ocean GO6 and GO7: a traceable hierarchy of model resolutions"

_Geoscientific Model Development, 2017_

## Referee Comment (RC1) · Anonymous Referee #1 · 30 Apr 2018

This is a generally well constructed and well written paper that describes newer versions of an important ocean climate model. Explanations of principal differences in state under forced integration are for the most part clear, along with the consideration of which changes in configuration are responsible. I have a number of mostly minor questions and points (some located by page and line number), below:

1. p. 4 l. 1: "This is to avoid instabilities associated with artificial cliffs in the bathymetry at the edge of the ice shelves where the ice cavities have been closed."
   As suggested by this comment, have you verified that GO7 is stable without partial-slip?

2. p. 4 l. 15: "with increased friction in the Indonesian Throughflow, Denmark Strait and Bab el Mandeb regions."
   Is this discussed in another paper? If so please include the citation. If not, can you make some brief comment on the basis for this change and the degree of impact?

3. p. 4 l. 16: Again, is there a reference to the impact of this scheme in Nemo? Based on the prominence of Overflows in Future Plans, one may infer that the scheme is not producing the desired result; a few words on this would be useful.

4. p. 3 l. 29:
   with the cube of the grid size ->
   with the cube of grid length

5. p. 7 l. 28: "in the winter"
   Perhaps this is fussy, but here and elsewhere it might be helpful to clarify as "austral winter" (and similarly for summer).

6. Section 4.4: With the factor of 2 reduction in isopycnal mixing coefficient, what happens with the Drake Passage transport?

7. Section 4.5: I would expect that Merino et al 2016 may be relevant (https://www.sciencedirect.com/science/article/pii/S1463500316300300), since it is on the NEMO-iceberg model and discusses impacts on sea ice?

8. p. 8 l. 28: "With less sea ice formation there was less sea ice overall and a widespread salinification of the surface waters in the Southern Ocean due to reduced melting offshore."
   This result is sufficiently counter-intuitive, given the tendency of drifting icebergs to deliver freshwater well offshore, that I would suggest including an explicit comment to the extent that reduced sea ice export dominates over iceberg export of freshwater, resulting in a net freshening of offshore waters.

And I admit, this is one place I'm asking myself: Really, are they sure that this is so generally true? Aren't there some places (iceberg-alleys) where the freshwater delivery of icebergs dominates over the reduced export of sea ice?

9. Section 4.6: Are you using the delta-Eddington shortwave scheme, which computes the albedos, or a simpler scheme in which 'base' albedos are parameters which are then adjusted based on temperature and thickness?

10. Section 5.2: is it not likely that the elimination of the Weddell Polynya is primarily due to the change in sea surface salinity (fresher in GO6, eliminating the salinity bias relative to EN4)?

11. p. 12 l. 8:
to indirectly salinify the surface layers ->
to indirectly make the surface layers more saline

12. p. 14 l. 16:
from Ronne Ice Shelf cavity, but are fresher... However, the overall the Filchner shelf ->
from the Ronne Ice Shelf cavity, but are fresher... However, overall the Filchner shelf

13. Fig. 3: could you include a few words here to indicate what this length scale $nn_h tauis$?

14. Fig. 7: this set of plots would be easier to assess if presented in units of m/yr (as in Marsh 2015).

15. Fig. 8:
fields are meaned over ->
means are taken over

16. Fig. 9 (and other similar opportunities): please use the figure caption to explain more than what one can already get from the annotations that are already embedded in the figure.

17. Fig. 17: what are the contour intervals? Does the interval change after 10?

18. Perhaps this would be addressed in final inspection before publication, but I will also make the point that it would be helpful to follow a policy of introducing acronyms.

---

## Referee Comment (RC2) · Anonymous Referee #2 · 13 May 2018

The paper describes several ocean configurations based on NEMO code and justifies the choice of the particular ones (GO6 and GO7) which will be used in the coupled version GC3.1 and in UKESM. The manuscript is well organized and consistently describes effects from the variety of options available in NEMO version 3.6 and below, providing sensitivity tests.

After reading the manuscript I first asked myself what I have learned. Besides the technical knowledge about all the details of the ocean component in GC3.1, some (already known) key aspects of improving the realism of the simulated ocean are provided. From the latter ones the main messages are:

1. increase in the near-surface mixing deepens the summer MLD and in the SH, through reduction in SST, suppresses the unrealistically large polynyas in the Wed-

dell Sea.

2. inclusion of multi-layer thermodynamics increases the summertime sea ice in the NH by ∼20% (although the authors attribute it to a slightly increase in p.9 l.22) as compared to zero-layer thermodynamics. Simultaneously, the effect in the SH is negligible.

3. Hollingsworth instability in the vector-invariant form of momentum advection needs a special treatment, otherwise it introduces large spurious tracer diffusivities.

I see this paper as a reference to GO6/GO7 configurations and the GMD is a proper journal for this. At the current stage some minor issues need to be solved before the manuscript can be published. Below I give my comments.

1. I could not find any information on performance and throughput. This section is definitely missing.

2. When giving sensitivity tests starting from figure 5 in section 4.4 I wonder about robustness of these tests. The coupled (15 years ?) as well as the forced (10 years) runs appear to be too short for making any conclusions. In the both runs I expect the differences to grow with time providing more insight into what is going on. If possible, I would recommend to integrate these tests up to at least 30 years long (even longer for coupled) and analyze the last decade like it was made in section 4.3.

3. Same is true for section 5.1. The CORE-II forced integration with $1/4°$ configuration shall be extended to at least one cycle of 60 years. Authors aim at using GO6 in CMIP6 and will probably do this run anyway.

4. Is Gent and McWilliams eddy parameterization used in GO6? If so it needs to be described in section 4.4 or in a separate section.

5. In section 6, besides what has been nicely discussed by authors, in fig. 18 I see that all Hovmöller diagrams for salinity show positive drift at and below 1000m which does not change with resolution. Does it come from Mediterranean? Having a longer run with $1/4°$ it would be nice to track the origin and the amplitude of this drift. Overall,

how do findings in this chapter compare to other CORE-II participants?

6. Section 5.2 Attributes changes in results to model changes only for SH. It would be great to track the changes in NH as well and, if applicable, track the key aspects I mentioned at the beginning.

7. Same is true for the summary. To my opinion it also requires more structure in summarising the main findings from chapters 4.x.

8. In the summary (P.15 L.8) as well as in abstract (P1 L.9) the authors attribute the improvements to the isopycnal mixing which I could not see in right panels of figures 5 and 6 (at least for the forced runs).

9. At the same time left panel of figure 5 where the difference from SST climatology is shown looks biased towards winter. Is it just a coincidence or something went wrong with averaging the model data?

10. P.3 L25 authors probably mean "Total Variation Diminishing"

---

## Author Comment (AC1) · 8 Jun 2018

We would like to thank the two reviewers for providing detailed and helpful reviews. We have responded to the individual points raised below. In addition to changes to the manuscript made in response to the reviewers' comments, we have also taken the opportunity to update and correct some of the references.

===================== Response to reviewer 1 =====================

1. p. 4 l. 1: "This is to avoid instabilities associated with artificial cliffs in the bathymetry at the edge of the ice shelves where the ice cavities have been closed." As suggested by this comment, have you verified that GO7 is stable without partial-slip?

The instabilities were seen in the coupled model and the partial-slip/no-slip condition

was applied in both the coupled and forced model for consistency. We don't yet have a coupled model with the ice shelf cavities open so we haven't tested to see if we can switch back to a free slip condition around Antarctica in this case. The forced tests of GO7 described in the paper used a no-slip condition along the Antarctic coastline, including along the front faces of the ice shelves. A free slip condition was applied at the grounding line within the ice shelf cavities.

2. p. 4 l. 15: "with increased friction in the Indonesian Throughflow, Denmark Strait and Bab el Mandeb regions." Is this discussed in another paper? If so please include the citation. If not, can you make some brief comment on the basis for this change and the degree of impact?

The UK "GO" configurations started off as a version of the Drakkar configuration as mentioned in Section 2. These spatial variations in the bottom friction coefficient were inherited as part of that configuration. The motivation for the increased friction in the Denmark Strait is to improve the dense water overflow (Gaelle Hervieux, Etude numerique des interactions courant-topographie : application au gyre subpolaire, auc seuils de Gibraltar et des mers Nordiques., Universite Joseph Fourier, Grenoble 1). For the Indonesian Throughflow and Bab al Mandeb strait we assume the motivation is to control the flow through the straits but we have been unable to trace references for these.

We have modified the third sentence in the introduction to Section 2 to clarify that the GO configurations are descended from a version of the Drakkar configuration.

3. p. 4 l. 16: Again, is there a reference to the impact of this scheme in Nemo? Based on the prominence of Overflows in Future Plans, one may infer that the scheme is not producing the desired result; a few words on this would be useful.

Unpublished experiments in idealised models performed by Tim Graham have indicated that the BBL scheme seems to have little impact in improving the poor representation of overflows in the partial-steps version of NEMO. This seems to be consistent

with the experiences of others in the NEMO community. Tim points out that there is a comment in the original Beckmann and Doscher paper to the effect that the scheme doesn't work very well with a stretched vertical grid of the sort that is standard in global ocean models. Apart from this we are unaware of any assessment of the efficacy of this scheme in global models in the peer-reviewed literature.

We have added a comment on this in Section 8 in the revised manuscript where we describe our plans to try to improve the overflow representation.

4. p. 3 l. 29: with the cube of the grid size -> with the cube of grid length

Changed in revised manuscript.

5. p. 7 l. 28: "in the winter" Perhaps this is fussy, but here and elsewhere it might be helpful to clarify as "austral winter" (and similarly for summer).

We have changed this instance and some other instances in the revised manuscript except where we thought it obvious from context, eg. "summertime mixed layers in the Arctic" probably doesn't need clarification.

6. Section 4.4: With the factor of 2 reduction in isopycnal mixing coefficient, what happens with the Drake Passage transport?

We have checked this in the extended 30 year integrations that we performed in response to the comments by reviewer 2. The answer is that the change to the isopycnal mixing coefficient has no significant impact on the Drake Passage transport in either the forced or the coupled test.

7. Section 4.5: I would expect that Merino et al 2016 may be relevant (https://www.sciencedirect.com/science/article/pii/S1463500316300300), since it is on the NEMO-iceberg model and discusses impacts on sea ice?

See response to point 8.

8. p. 8 l. 28: "With less sea ice formation there was less sea ice overall and a

widespread salinification of the surface waters in the Southern Ocean due to reduced melting offshore." This result is sufficiently counter-intuitive, given the tendency of drifting icebergs to deliver freshwater well offshore, that I would suggest including an explicit comment to the extent that reduced sea ice export dominates over iceberg export of freshwater, resulting in a net freshening of offshore waters. And I admit, this is one place I'm asking myself: Really, are they sure that this is so generally true? Aren't there some places (iceberg-alleys) where the freshwater delivery of icebergs dominates over the reduced export of sea ice?

Merino et al find a general freshening of the surface waters and an increase in sea ice due to the inclusion of the iceberg model, which is the opposite of the results that we describe. The key difference is that in our control experiment (and in the control experiment of Marsh et al), a representation of iceberg meltwater was put in close to the coast of Antarctica. This created a thin layer of fresh water and stimulated the formation of sea ice near to the coast. When we spread the iceberg melting over a larger area using the iceberg model we remove this intense freshening of coastal waters so the sea ice formation rate drops and there is less sea ice overall. In the control experiment in Merino et al there is no representation of iceberg meltwater, so the addition of the iceberg model gives a general surface freshening and an increase in the sea ice.

We have included a reference to Merino et al 2016 in Section 4.5 of the revised paper and included a discussion of the contrast between their results and our results.

9. Section 4.6: Are you using the delta-Eddington shortwave scheme, which computes the albedos, or a simpler scheme in which "base" albedos are parameters which are then adjusted based on temperature and thickness?

The shortwave scheme is the CCSM3 scheme in CICE which a relatively simple scheme, not the delta-Eddington scheme.

We have added a description of the albedo scheme in Section 2.5. We have also

clarified the wording in Section 4.6.

10. Section 5.2: is it not likely that the elimination of the Weddell Polynya is primarily due to the change in sea surface salinity (fresher in GO6, eliminating the salinity bias relative to EN4)?

Examination of the impact on the sea ice of the reduction in the isopycnal mixing coefficient (which is primarily responsible for the near-surface freshening) shows that this does cause a year-round increase in the total sea ice volume in the Southern Hemisphere. However the main increase in sea ice volume is between the Antarctic Peninsula and the Ross Sea rather than in the Weddell Sea. Furthermore, the increase in overall sea ice volume is significantly smaller in this test than in the test of the vertical mixing (nn_htau) change. So we conclude that the vertical mixing change is the most significant factor for the Weddell Sea polynyas closing (consistent with the Heuze et al and Kjellsson et al results).

We have added an extra plot to Fig 4 to show the impact of reducing the isopycnal diffusion on the sea ice volume and added text to Section 4.4 referring to this.

11. p. 12 l. 8: to indirectly salinify the surface layers -> to indirectly make the surface layers more saline

Changed in revised manuscript.

12. p. 14 l. 16: from Ronne Ice Shelf cavity, but are fresher. . . However, the overall the Filchner shelf -> from the Ronne Ice Shelf cavity, but are fresher. . . However, overall the Filchner shelf

Changed in revised manuscript.

13. Fig. 3: could you include a few words here to indicate what this length scale nn_htau is?

Description added in revised manuscript.

14. Fig. 7: this set of plots would be easier to assess if presented in units of m/yr (as in Marsh 2015).

The plots in Figure 7 have been replotted with units of m/yr.

15. Fig. 8: fields are meaned over -> means are taken over

Changed in revised manuscript.

16. Fig. 9 (and other similar opportunities): please use the figure caption to explain more than what one can already get from the annotations that are already embedded in the figure.

We have adjusted most of the figure captions in the revised manuscript to avoid repeating information in the individual panel titles.

17. Fig. 17: what are the contour intervals? Does the interval change after 10?

The colours are plotted at 1 Sv intervals and the line contours plotted at 2 Sv intervals. The colours saturate beyond the limits of the colour bar.

We have specified the line contour intervals in the figure caption in the revised manuscript.

18. Perhaps this would be addressed in final inspection before publication, but I will also make the point that it would be helpful to follow a policy of introducing acronyms.

We have tried to make sure that all acronyms are expanded (often in footnotes) in the revised manuscript.

==================== Response to reviewer 2 =====================

1. I could not find any information on performance and throughput. This section is definitely missing.

We have added a new Section 2.6 with some model performance information in the revised manuscript.

2. When giving sensitivity tests starting from figure 5 in section 4.4 I wonder about robustness of these tests. The coupled (15 years ?) as well as the forced (10 years) runs appear to be too short for making any conclusions. In the both runs I expect the differences to grow with time providing more insight into what is going on. If possible, I would recommend to integrate these tests up to at least 30 years long (even longer for coupled) and analyze the last decade like it was made in section 4.3.

For the sensitivity integrations presented in Section 4.4 we have extended the forced and coupled integrations to 30 years as suggested by the reviewer. The conclusions for the sensitivities of the near-surface temperature and salinity fields at high latitidues are unchanged (but see also response to reviewer's point 9).

Figures 5 and 6 changed to show results from 30-year integrations and text in Section 4.4 modified accordingly.

3. Same is true for section 5.1. The CORE-II forced integration with 1/4? configuration shall be extended to at least one cycle of 60 years. Authors aim at using GO6 in CMIP6 and will probably do this run anyway.

We accept that the 30 year integrations presented here would be too short for a full assessment of the long-term behaviour of the GO6 configurations. However the two main areas of focus of the paper are to assess updates to the near-surface mixing and surface processes compared to GO5 and to establish a traceable hierarchy of resolutions. We would argue that the integrations presented here are sufficient to assess the upper ocean sensitivities, which equilibrate on relatively short timescales. Where we look at deeper processes, such at the overturning circulation in Section 6 we are careful not to make any claims about long-term behaviour. The GO6 configuration is being run for the full CORE2 protocol and we plan to present a full analysis of these integrations in a future paper.

We have added text to this effect at the beginning of Section 5.1 in the revised manuscript.

[Figure]

4. Is Gent and McWilliams eddy parameterization used in GO6? If so it needs to be described in section 4.4 or in a separate section.

A version of the Gent-McWilliams eddy parametrisation is only used in the ORCA1 model (mentioned in Section 2.3 - reference to Held and Larichev paper).

5. In section 6, besides what has been nicely discussed by authors, in fig. 18 I see that all Hovmöller diagrams for salinity show positive drift at and below 1000m which does not change with resolution. Does it come from Mediterranean? Having a longer run with 1/4? it would be nice to track the origin and the amplitude of this drift. Overall, how do findings in this chapter compare to other CORE-II participants?

The positive drift at depth in the salinity field in all three resolutions appears to be associated with a positive drift in the salinity of the Mediterranean Outflow water in the north-east Atlantic. We hope to say more about the long term drifts and the comparison with other CORE-II participants in our future paper describing the CORE-II GO6 integrations.

We have added a sentence about the positive salinity drift at depth in Section 6.

6. Section 5.2 Attributes changes in results to model changes only for SH. It would be great to track the changes in NH as well and, if applicable, track the key aspects I mentioned at the beginning.

The main focus of the model changes between GO5 and GO6 was on the Southern Ocean hence the focus in the paper. But we have added a brief description and attribution of northern hemisphere changes in Sections 5.1 and 5.2 and in the Summary (Section 8) in the revised manuscript.

7. Same is true for the summary. To my opinion it also requires more structure in summarising the main findings from chapters 4.x.

We have reworded the Summary in the revised manuscript to give a more detailed summary of the GO5-GO6 changes and attribution.

8. In the summary (P.15 L.8) as well as in abstract (P1 L.9) the authors attribute the improvements to the isopycnal mixing which I could not see in right panels of figures 5 and 6 (at least for the forced runs).

We attribute the reduction of the salty bias in the near surface salinity in the Southern Ocean to the reduction in the isopycnal diffusion coefficient as described in Section 5.2 (p12 l 22-30) and as seen in the surface freshening in Fig 6d. The improvements in the Southern Ocean temperature and sea ice simulation are mainly attributable to the change in the near-surface mixing coefficient (nn_htau) (Section 5.2 second paragraph).

We have reworded the summary (Section 8) in the revised manuscript to make this clearer.

9. At the same time left panel of Figure 5 where the difference from SST climatology is shown looks biased towards winter. Is it just a coincidence or something went wrong with averaging the model data?

In the process of extending the sensitivity integrations for Section 4.4 we discovered a mistake in the coupled test whereby the near-surface vertical mixing parameter nn_htau had been changed as well as the isopycnal diffusion coefficient. This accounts for the north-south hemispheric asymmetry noted by the reviewer in the coupled model plots in Fig 5. We have done a new 30-year test for the coupled model and this asymmetry is no longer apparent.

10. P.3 L25 authors probably mean "Total Variation Diminishing".

Changed in revised manuscript.

———————————————————